# The immune receptor SLAMF5 regulates myeloid-cell mediated neuroinflammation in multiple sclerosis

Laura Bellassen[1], Keren David[1], Bar Lampert[1], Avital Sarusi-Portuguez[2], Michael Tsoory[3], Jazz Lubliner[4], Eran Hornstein[4], Michael Osherov[5], Ron Milo[5], Ori Brenner[6], Shirly Becker-Herman[1], Idit Shachar [1]*

1 Department of Systems Immunology, Weizmann Institute of Science, Rehovot, Israel, 2 The Mantoux Bioinformatics Institute of the Nancy and Stephen Grand Israel National Center for Personalized Medicine, Weizmann Institute of Science, Rehovot, Israel, 3 Behavioral and Physiological Phenotyping Unit, Weizmann Institute of Science, Rehovot, Israel, 4 Department of Molecular Genetics, Weizmann Institute of Science, Rehovot, Israel, 5 Department of Neurology, Barzilai University Medical Center, Ashkelon, Israel, 6 Department of Veterinary Resources, Weizmann Institute of Science, Rehovot, Israel

* idit.shachar@weizmann.ac.il

## Abstract

Multiple sclerosis (MS) is a chronic neurological disorder characterized by demyelination of the central nervous system (CNS), leading to a broad spectrum of physical and cognitive impairments. Myeloid cells within the CNS, including microglia and border-associated macrophages, play a central role in the neuroinflammatory processes associated with MS. Activation of these cells contributes to the local inflammatory response and promotes the recruitment of additional immune cells into the CNS. SLAMF5 is a cell surface receptor that functions as a homophilic adhesion molecule, capable of modulating immune cell activity through both activating and inhibitory signals. In this study, we investigated the expression and function of SLAMF5 in CNS-resident and peripheral myeloid cells using the murine model of MS, experimental autoimmune encephalomyelitis (EAE). Our findings demonstrate that both total and brain-specific SLAMF5 deficiency in myeloid cells leads to decreased expression of activation and costimulatory molecules, including MHC class II (MHCII) and CD80. This downregulation is mediated, at least in part, through the transcription factor BHLHE40 and its regulation of CD52, resulting in delayed onset and reduced progression of the disease. Furthermore, pharmacological blockade of SLAMF5 in the brain halted disease progression and reduced the expression of myeloid activation markers. In human studies, SLAMF5 blockade in peripheral monocytes from MS patients and in induced pluripotent stem cell (iPSC)-derived microglia reduced the expression of HLA-DR, CD80, and CD52. Together, these results identify SLAMF5 as a key regulator of myeloid cell activation in neuroinflammation and suggest that it may represent a promising therapeutic target for autoimmune disorders such as MS.

**Data availability statement:** The fsc files have been deposited in Zenodo https://doi.org/10.5281/zenodo.16909365. RNA-seq data have been deposited in GEO under accession number GSE306091.

**Funding:** This work was supported by the Briskin Foundation (IS). The funder had no role in study design, data collection and analysis, decision to publish, or preparation of the manuscript.

**Competing interests:** The authors have declared that no competing interests exist.

**Abbreviations:** AUC, Area Under the Curve; BAMs, border-associated macrophages; BBB, brain–blood barrier; cKO, conditional knockout mice; CLL, chronic lymphocytic leukemia; CNS, central nervous system; EAE, experimental autoimmune encephalomyelitis; FACS, flow cytometry; FBS, fetal bovine serum; GPI, glycosylphosphatidylinositol; H&E, hematoxylin and eosin; IACUC, Institutional Animal Care and Use Committee; ICV, intracerebroventricular; IP, intraperitoneally; iPSC, induced pluripotent stem cell; IL-1β, interleukin-1 beta; I.V., intravenous; LPS, lipopolysaccharide; MHCII, MHC class II; MOG, myelin oligodendrocyte glycoprotein; MS, multiple sclerosis; PBMC, peripheral blood mononuclear cell; PBS, phosphate-buffered saline; PCA, Principal Component Analysis; RNA-seq, RNA sequencing; SLAM, Signaling Lymphocytic Activation Molecule; TAM, tamoxifen; TNF-α, tumor necrosis factor-alpha; WT, wild type.

## Introduction

Multiple sclerosis (MS) is the most common chronic inflammatory autoimmune disease of the central nervous system (CNS) and the leading cause of non-traumatic disability in young adults, particularly young women [1,2]. It is characterized by chronic demyelination, oligodendrocyte loss, and progressive axonal and neuronal degeneration [3,4]. The clinical presentation and progression of MS are highly heterogeneous, reflecting variability in underlying pathological mechanisms [5]. MS triggers inflammation in both white and gray matter, driven by the infiltration of immune cells, primarily CD8+ T cells, CD4+ T cells, and activated macrophages, which initiate a damaging inflammatory cascade within the CNS [6]. This response is further amplified by the accumulation of inflammatory cytokines, reactive oxygen species, and free metal ions released by infiltrating immune cells [7,8]. These factors collectively contribute to neuronal injury and activate resident brain immune cells, such as microglia and astrocytes [9,10]. A breakdown in immune tolerance, often due to imbalances in the number and/or function of specific T cell subsets, is believed to initiate these pathological events [11].

The myeloid cell population in the brain is heterogeneous, consisting primarily of microglia, the resident innate immune cells of the CNS [12]. Microglia, which make up approximately 5%–20% of CNS cells, are characterized by the expression of macrophage-associated markers such as CD11b [12,13]. They originate from yolk sac macrophages that migrate into the CNS during early embryonic development [14]. As mononuclear phagocytic cells, microglia play a vital role in maintaining CNS homeostasis. They are involved in a wide range of functions, including pathogen clearance, neuronal support, myelin turnover, and the pruning of excess synaptic connections [15,16].

Border-associated macrophages (BAMs) are a distinct subset of macrophages located at the interface between the CNS and peripheral tissues, including the meninges [17]. In the context of MS, BAMs become activated and contribute to the inflammatory milieu within the CNS. Their presence in MS lesions suggests they may play a role in recruiting additional immune cells to sites of injury.

Moreover, BAMs secrete pro-inflammatory cytokines such as tumor necrosis factor-alpha (TNF-α) and interleukin-1 beta (IL-1β), which are implicated in myelin degradation and lesion formation [18].

As the brain communicates with the periphery through the brain–blood barrier (BBB), infiltrating macrophages and monocytes that enter the CNS in response to inflammation play a critical role in the development and progression of MS. These cells contribute to disease pathology through antigen presentation, amplification of inflammatory signals, and modulation of the adaptive immune response, functions that they share with resident microglia [19,20]. Importantly, the functional state of these cells, whether activated, senescent, or otherwise, significantly influences their behavior and impact in the context of MS [21].

Experimental autoimmune encephalomyelitis (EAE) is the most widely used murine model for studying MS. EAE mimics key pathological features of MS, including chronic, non-relapsing disease progression, mononuclear inflammatory infiltration,

and demyelination [22,23]. The disease is typically induced by immunizing mice with myelin-derived antigens, such as myelin oligodendrocyte glycoprotein (MOG) [24,25] and is primarily mediated by Th1 and Th17 CD4$^+$ T cells [26]. During disease progression, activated microglia and T cells engage in both direct cell-to-cell interactions and cytokine-mediated signaling, driving neuroinflammation. The extent of myelin damage is positively correlated with the abundance of T cells and activated microglia in both MS and the EAE model [27,28].

The SLAM (Signaling Lymphocytic Activation Molecule) family of immunomodulatory receptors mediates interactions between immune cells and their microenvironment [29]. This family comprises nine members that are differentially expressed, primarily on immune cells. One key member, SLAMF5 (also known as CD84), is broadly expressed on B cells, T cells, monocytes, macrophages, dendritic cells, and natural killer cells. SLAMF5 functions as a homophilic adhesion molecule whose signaling can either activate or inhibit leukocyte activity, depending on the context [30,31]. The downstream effects of SLAM receptors are predominantly mediated by SLAM-associated protein family adapters. Dysregulation of SLAMF5 signaling has been linked to various autoimmune and lymphoproliferative disorders [32].

Previous studies have shown that SLAMF5 facilitates interactions between chronic lymphocytic leukemia (CLL) cells and their microenvironment, promoting CLL cell survival and suppressing T cell activity through upregulation of PD-L1 [33–35]. Moreover, SLAMF5 expression in regulatory B cells (Bregs) has been shown to modulate autoimmune responses in the EAE model. This effect is mediated by the IL-10-producing Breg population and the transcription factor c-Maf, contributing to regulation of disease severity [36].

In the current study, we investigated the role of SLAMF5 in the development of EAE by examining its expression on myeloid cell populations in both the brain and peripheral tissues of EAE-induced mice, as well as in the peripheral blood of MS patients and in induced pluripotent stem cell (iPSC)-derived microglia.

Our findings demonstrate that disrupting SLAMF5-mediated cell–cell interactions—either through genetic deficiency or receptor blockade—attenuates the inflammatory phenotype of brain myeloid cells. This effect is associated with downregulation of the transcription factor BHLHE40, which controls the expression of CD52, a glycosylphosphatidylinositol-anchored surface glycoprotein whose function in myeloid cells remains poorly understood [37].

These results suggest that SLAMF5 may serve as a potential therapeutic target to mitigate neuroinflammation driven by myeloid cells in MS.

## Materials and methods

### Ethics statement

All animal procedures were reviewed and approved by the Institutional Animal Care and Use Committee (IACUC) of Weizmann Institute of Science. All experiments were conducted in accordance with institutional and national guidelines for the care and use of laboratory animals. **The IACUC number is 03390422-2.**

### Mice

C57BL/6 mice (WT), SLAMF5 deficient mice, CX3CR1$^{creER}$ (Strain #:020940; The Jackson Laboratory) SLAMF5-flox (C57BL/6JGpt-Cd84em1Cflox/Gpt; Strain#:T020035; Gempharmatech), CX3CR1$^{creER}$ negative SLAMF5-flox were used. The Cre-flox mice were induced with 20 mg/mL of tamoxifen (Sigma-Aldrich) resuspended in corn Oil (Sigma-Aldrich) injected intraperitoneally (IP) for five consecutive days. All animals used in this study were between 9 and 14 weeks of age. Both male and female mice were included, and experimental groups were age- and sex-matched for each experiment.

### Induction of experimental autoimmune encephalomyelitis (EAE)

EAE was induced as previously described [38]. Briefly, mice were injected subcutaneously with 100 µg of MOG$_{35–55}$ peptide (synthesized by Genscript) emulsified in incomplete Freund's adjuvant supplemented with 3 mg/mL heat-inactivated

*Mycobacterium tuberculosis* (Sigma-Aldrich). Pertussis toxin (90 ng/mouse; Sigma-Aldrich) was administered IP immediately following the $MOG_{35-55}$ injection and again 48 hours later.

Starting on day 9 postimmunization, mice were monitored daily and scored for clinical signs of EAE using the following scale:

0—no disease; 1—loss of tail tone; 2—hind limb weakness; 3—complete hind limb paralysis; 3.5—hind limb paralysis with partial forelimb involvement; 4—complete paralysis of all limbs; 5—moribund or death.

Following institutional IACUC guidelines, mice were sacrificed if their clinical EAE score reached 3 or if they experienced a weight loss exceeding 20%.

## Preparation of brain murine cells

Following euthanasia and perfusion with cold phosphate-buffered saline (PBS; Sartorius), murine brains were dissected and collected in ice cold PBS. Tissues were minced and incubated for 20 min at 37°C in 1 mL of Hanks' Balanced Salt Solution (HBSS; Sigma-Aldrich) containing 2% bovine serum albumin (BSA; MP Biomedicals), 1 mg/mL collagenase D (Sigma-Aldrich), and 50 µg/mL DNase I (Sigma-Aldrich).

Following enzymatic digestion, brain homogenates were passed through a 70 µm cell strainer (SPL Life Sciences), washed with cold FACS buffer (2% fetal bovine serum (FBS; Sigma-Aldrich), 1 mM EDTA in PBS), and centrifuged at 4°C, 900g for 5 min. The supernatant was discarded, and the pellet was resuspended in 3 mL of 40% Percoll (Cytiva) for immune cell enrichment. Samples were centrifuged without acceleration or brake at 25°C, 900g for 15 min. The resulting cell pellet was resuspended in FACS buffer, filtered through an 80 µm mesh (SPL Life Sciences), washed with 5 mL of FACS buffer, and centrifuged at 4°C, 400g for 5 min. Cells were then stained with antibodies and analyzed by flow cytometry (FACS) [27]. A full list of antibodies is provided in S1 Table.

## Splenocyte preparation

Spleens were dissected and collected in PBS supplemented with 2% FBS. Tissues were homogenized through a 100 µm cell strainer to generate a single-cell suspension, followed by red blood cell lysis using Red Blood Cell Lysis Buffer (1.55 M NH4C, 0.1 M KHCO3, 1 mM EDTA, ddH2O). After lysis, cells were washed with PBS and passed through a 40 µm cell strainer (SPL Life Sciences) to remove debris.

Cells were then stained with antibodies and analyzed by FACS. The list of antibodies used is provided in S1 Table.

## Human peripheral blood lymphocyte preparation (PBMC)

Peripheral blood samples from MS patients and age- and sex-matched healthy controls were obtained in accordance with the Institutional Review Board of Barzilai Medical Center (Ashkelon, Israel). Informed consent was obtained in writing from all participants.

Whole blood was centrifuged at 2,700 RPM for 10 min at room temperature. The upper plasma and lower red blood cell fractions were discarded, and the remaining leukocyte layer was collected. Leukocytes were layered over Ficoll-Paque (Sigma-Aldrich) and centrifuged at 1,400 RPM for 20 min without brakes. The peripheral blood mononuclear cell (PBMC) layer was carefully collected.

PBMCs were then incubated with 60 µg/mL SLAMF5-blocking antibody or an isotype control antibody (Ultra-LEAF purified mouse IgG2a; BioLegend) for 48 h. Following treatment, cells were stained with antibodies and analyzed by FACS. The complete list of antibodies is provided in S2 Table.

## Human induced pluripotent stem cells-derived microglia (iPSC)

Human iPSCs from the Knockout Laboratory (KOLF2.1J, wild type (WT)) were differentiated into microglia-like cells [39–41]. iPSCs were seeded into 96-well suspension plates in mTeSR1 medium supplemented with 50 ng/

mL rhBMP4 (Peprotech, 314-BP), 50 ng/mL VEGF (Peprotech, 100-20), 20 ng/mL SCF (Peprotech, 300-07) and 10 nM Y-27632 dihydrochloride (Selleck Chemicals) to induce embryoid bodies. Each day, half of the medium was removed, and an equal volume of fresh medium was added. After 4 d, 12 embryoid bodies were transferred into each well of a six-well plate in X-VIVO 15 (Lonza, BE02-060Q) containing 100 ng/mL M-CSF (Peprotech, 300-25), 25 ng/mL IL-3 (Peprotech, 200-03), 2 mM GlutaMAX, 55 µM 2-mercaptoethanol (Gibco, 31350-10), and 100 U/ mL penicillin-streptomycin (Biological Industries, 03-031-1B). iPSC-derived progenitor microglia (ipMGs) were collected weekly from the supernatant and were incubated with 10 ng/ mL of IL-34 (Peprotech, 200-34). The cells collected were seeded and matured for an additional 2 weeks with ITMG media [42]. Half of the media was changed every 3–4 days. The cells were treated with 10 ng/mL Lipopolysaccharide (LPS (L2880, Sigma-Aldrich)) to induce a pro-inflammatory environment. After 48h of LPS treatment, the cells were incubated with 60 µg/mL SLAMF5 blocking antibody or the Isotype control antibody (Biolegend) for 48 h. After treatment, the cells were labeled with antibodies and analyzed using FACS. Antibodies are listed in S2 Table. A total of three independent experiments were performed: the first experiment was conducted using initial batch of iPSCs, and the subsequent two experiments were performed using a second batch of cells.

### Preparation of histological samples and scoring

Mice were sacrificed by an overdose of $CO_2$. The head and vertebral columns were dissected and fixed in 4% neutral buffered formalin for several days. Following decalcification (Surgipath Decalcifier I, Leica Biosystems) the caudal skull was cut into four transverse slices with the brain in situ. Three to four transverse sections were collected from the cervical, thoracic, and lumbosarcal segments of the vertebral column with the spinal cord in situ. In addition, a longitudinal section was collected from the thoracic region. The samples were processed routinely, embedded in paraffin, sectioned at 4–5 µm and stained with hematoxylin and eosin (H&E).

Semi-quantitative microscopic evaluation of the inflammation and the extent of spinal cord damage was performed on H&E-stained sections by an expert pathologist, using a score of 0–4 (0—normal, 1—minimal, 2—ild, 3—moderate, and 4—severe).

### Intravenous (I.V.) injection of SLAMF5 blocking antibody for in vivo treatment

To evaluate the role of SLAMF5 in disease progression, EAE induced mice were treated with SLAMF5 blocking antibody or isotype-matched control antibody. Each mouse received 150 µg of antibody diluted in 200 µL of sterile PBS, administered I.V. via the tail vein. Antibody injections were carried out on days 7, 9, and 12 postEAE induction.

### In vivo treatment by intracerebroventricular (ICV) injection of SLAMF5 blocking antibody

EAE was induced in WT mice as described previously. On day 5 postEAE induction, mice underwent intracerebroventricular (ICV) injection of SLAMF5 antibody or IgG control via stereotaxic surgery to assess the central effects of SLAMF5 blockade. Stereotaxic injections were performed under general anesthesia using aerosolized isoflurane (induction: 4%; maintenance: 2%–1.5%) delivered via a precision vaporizer system. Mice were positioned in a stereotaxic frame, and bilateral injections were carried out at the following coordinates relative to Bregma: anteroposterior (AP): −0.6 mm, mediolateral (ML): ±1.1 mm, and dorsoventral (DV): −2.0 mm. A volume of 2 µL containing 3 µg of the respective antibody was slowly injected into each lateral ventricle using a Hamilton syringe equipped with a fine-gauge needle to minimize tissue disruption.

A comprehensive pain management protocol was implemented to minimize animal distress and ensure ethical compliance. Preoperative analgesia included an I.P. injection of Buprenorphine (0.05 mg/kg) administered 30 min prior to surgery. Local anesthesia was provided by applying lidocaine cream to the scalp before incision. Postoperative analgesia consisted of Carprofen (5 mg/kg, I.P.) administered once daily for three consecutive days following surgery.

PLOS Biology

Following ICV injection, mice were monitored daily for clinical signs of EAE and overall health status until day 14 postinduction. On day 14, mice were euthanized. Brain myeloid cells and splenocytes were isolated as described above. Single-cell suspensions were stained with antibodies and analyzed using FACS. Antibodies are listed in S1 Table.

### Flow cytometry and staining

Cells were stained using specific antibodies, as previously described [34].

FACS analysis was performed using FACS Canto (BD Biosciences) and data were collected using FACSDIva8 (BD Biosciences). FACS data analysis was done using Flowjo v10. Antibodies are listed in S1 and S2 Tables.

### RNA sequencing and analysis

EAE was induced in WT and SLAMF5-deficient mice. At the peak of disease, mice were sacrificed, and their brains were harvested. Brain tissues were processed, and immune cells were isolated and stained with viability dye, anti-CD45, and anti-CD11b antibodies. The CD45⁺CD11b⁺ population was sorted using a BD FACSMelody cell sorter. RNA was then extracted from the sorted cells and subjected to downstream gene expression analysis. RNA-seq libraries were prepared at the Crown Genomics institute of the Nancy and Stephen Grand Israel National Center for Personalized Medicine, Weizmann Institute of Science. A bulk adaptation of the MARS-Seq protocol [43,44] was used to generate RNA-Seq libraries for expression profiling of SLAMF5. Briefly, 60 ng of input RNA from each sample was barcoded during reverse transcription and pooled. Following Agencourct Ampure XP beads cleanup (Beckman Coulter), the pooled samples underwent second-strand synthesis and were linearly amplified by T7 in vitro transcription. The resulting RNA was fragmented and converted into a sequencing-ready library by tagging the samples with Illumina sequences during ligation, RT, and PCR. Libraries were quantified by Qubit and Tape Station as previously described [43,44]. Sequencing was done on a Nextseq 75 cycles high output kit, allocating 20M reads per sample (Illumina; single read sequencing).

Poly-A/T stretches and Illumina adapters were trimmed from the reads using cutadapt; resulting reads shorter than 30 bp were discarded. The remaining reads were mapped onto 3′ UTR regions (1,000 bases) of the M. musculus, GRCm39 genome according to Refseq annotations, using STAR [45], with the End-to-end option and outFilterMismatchNoverLmax was set to 0.05. Deduplication was carried out by flagging all reads that were mapped to the same gene and had the same UMI. Counts for each gene were quantified using htseq-count [46], using the gtf above. Only uniquely mapped reads were used to determine the number of reads that map to each gene (intersection-strict mode). UMI counts were corrected for saturation by considering the expected number of unique elements when sampling without replacement. Differential analysis was performed using DESeq2 package [47] with the betaPrior, cooksCutoff, and independent filtering parameters set to False. Raw $P$ values were adjusted for multiple testing using the procedure of Benjamini and Hochberg. Differentially expressed genes were determined by a $p$-adj of < 0.05, absolute fold changes >1.5, and a count of at least 30 at least in one sample. Heatmap plotting was done using the Complex Heatmap package from R using log 2 DESeq2 normalized counts of each gene that have been centered to have mean zero.

### Chip-qPCR

Chromatin immunoprecipitation was conducted as described before [48,49]. Immune cells were purified from brains derived from EAE-induced mice at the peak of the disease. The cells were stained for dead cells, CD45 and CD11b. The double positive population was sorted using BD FACS Melody. The cells were incubated with 100 µg SLAMF5 blocking antibody or with the Isotype control antibody for 1 h, then cross-linked with DSG (disuccinimidyl glutarate) and fixed. BHLHE40-bound DNA fragments were precipitated, and ChIP-DNA was processed. The samples were analyzed by qPCR for CD52, TMEM119, and IL10RA promotor areas. The primers are provided in S3 Table.

### Real-time PCR and primers

RNeasy Mini kit was used (Qiagen) to extract RNA from mice and cell lines samples. To synthesize cDNA, a Reverse Transcription kit was used (Qiagen). Real-time PCR was performed using a Light-Cycler 480 Instrument (Roche Diagnostics) and Light-Cycler 480 SYBR Green Master (Roche Diagnostics). The primer list is provided in S3 Table.

### Separation of CD11b cells

Brain and murine spleens were dissected postmortem. After collecting the cells from the brain and spleen (see above) of EAE-induced mice, CD11b microbeads (Miltenyi Biotec) were added to the cells and separated on MS columns by magnetic separation (Miltenyi Biotec). The CD11b positive cells were incubated with 60 µg/mL of the SLAMF5 blocking antibody or isotype control antibody for 48h.

### Cell line and media

The murine microglial cell line, N9, was cultured in RPMI 1640 with 10% of FBS and 1% of Penicillin-Streptomycin (Biological Industries). To induce proinflammatory stimuli, 10 µg LPS (L2880, Sigma-Aldrich) was added for 3 days. The SLAMF5 blocking antibody, or the isotype control antibody Ultra-LEAF Purified Mouse IgG2a (Biolegend) antibody were added, after the 3 days LPS treatment, for 24 h at a concentration of 60 µg/m.

LPS stimulation can effectively mimic M1 polarization—a phenotype commonly observed in neuroinflammatory diseases like EAE. Specifically, LPS- induced activation leads to upregulation of MHC class II (MHCII) and iNOS, as well as downregulation of markers such as arginase 1, FIZZ1, and CX3CR1, which are indicative of M1 polarization [50].

### SiRNA

siRNA was introduced by electroporation using a Nepagene (Ichikawa, Chiba, Japan) Super Electroporator NEPA21 Type II. N9 cells were electroporated with silencing sequences including: 5nmol Silencer pre-designed Cd52 siRNA (Thermofisher), 5nmol Silencer pre-designed BHLHE40 siRNA (Thermofisher), and 5nmol Silencer Negative Control siRNA (Thermofisher). Twenty-four hours after the transfection, 10 µg LPS (L2880, Sigma-Aldrich) was added to induce a pro-inflammatory environment.

### Statistics

Statistical analysis was performed using **GraphPad** Prism 8 software (San Diego, California, USA).

### Model figure

The Schematic figures and the model were created with BioRender.com.

## Results

### SLAMF5 regulates myeloid cell activation in the brain of EAE-induced mice

The role of SLAMF5 in regulating myeloid cell activity during neuroinflammation remains largely unknown. To investigate whether SLAMF5 expression on myeloid cells influences the development of EAE, WT and SLAMF5-deficient (SLAMF5$^{-/-}$) mice were immunized with the MOG$_{35-55}$ peptide, and disease progression was monitored over 15 days using a standard EAE scoring system (see Materials and methods). Consistent with previous findings [36], SLAMF5$^{-/-}$ mice exhibited delayed disease onset and reduced motor dysfunction compared to WT controls (see S1A and S1B Fig).

Analysis of SLAMF5 expression on total brain myeloid cells (CD45$^+$CD11b$^+$) [51] revealed significantly higher expression levels in EAE-induced mice compared to healthy controls (Fig 1A and 1B). Moreover, SLAMF5 expression positively

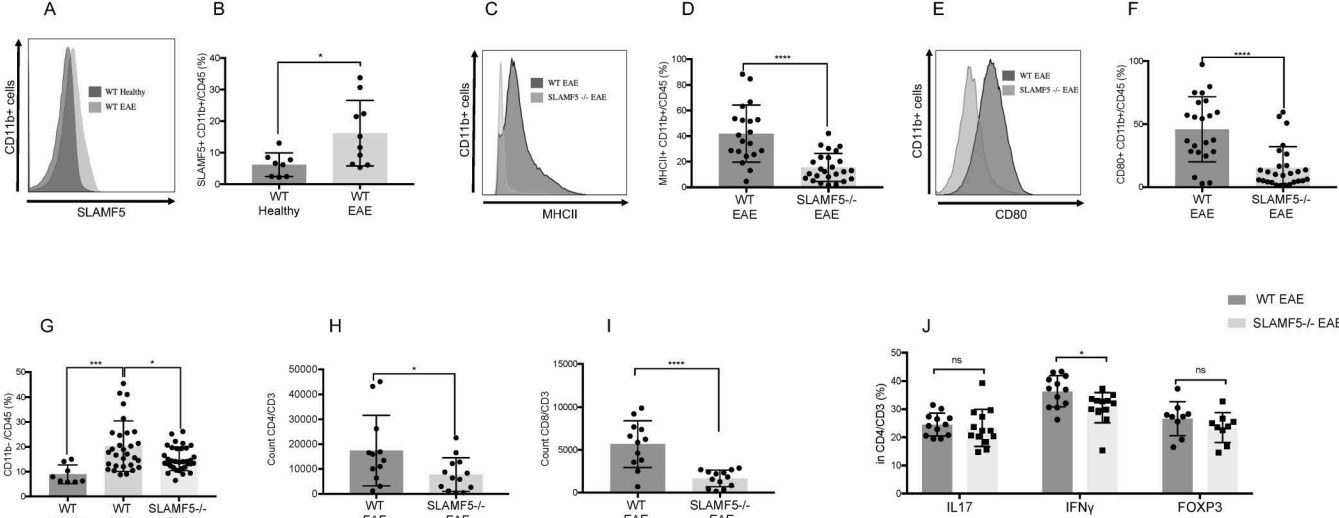

**Fig 1. SLAMF5 regulates brain myeloid cell activation in EAE induced mice.** EAE (MOG35-55) was induced in WT and in SLAMF5-deficient mice. Mice were sacrificed at the peak of the disease. Dead cells were excluded from analysis by Zombie Live/Dead staining. **(A, B)** Analysis of SLAMF5 expression in brain myeloid cells (CD45+CD11b+) of WT Healthy and WT EAE induced mice. **(A)** Representative histogram, and **(B)** bar chart show SLAMF5 expression in % in brain myeloid population. (WT Healthy $n = 8$; WT EAE $n = 10$). Graph shows three independent determinations. **(C, D)** Analysis of MHCII expression in brain myeloid cells (CD45+CD11b+) of WT and SLAMF5−/− EAE induced mice. Bar graph shows the expression of MHCII in % in myeloid cells (WT $n = 20$; SLAMF5−/− $n = 25$). **(E, F)** Analysis of CD80 expression in brain myeloid cells (CD45+ CD11b+) of WT and SLAMF5-deficient EAE-induced mice. Bar graph shows the expression of CD80 in % in myeloid cells (WT $n = 22$; SLAMF5−/− $n = 27$). Graph shows eight independent determinations. Two-tailed unpaired Student $t$ test with 95% confidence levels. **(G)** The percentage of CD11b− cells of total CD45+ immune cells in WT Healthy, WT, and SLAMF5−/− EAE induced mice. (WT Healthy $n = 8$; WT EAE $n = 30$; SLAMF5−/− $n = 34$). Graph shows eight independent repeats. Ordinary one-way ANOVA with Dunnett multiple comparison tests. **(H)** Count of CD3 CD4 positive cells in the brains of WT and SLAMF5−/− EAE induced mice. **(I)** Count of CD3 and CD8 positive cells in the brain of WT and SLAMF5−/− EAE induced mice. **(J)** Expression of IL-17, Interferon γ and FOXP3 in % in CD3 CD4 positive T cells (WT EAE, $n = 12$; SLAMF5, $n = 13$). Graphs show three independent determinations. Two-tailed unpaired Student $t$ test with 95% confidence levels (*$P < 0.05$, **$P < 0.01$, ***$P < 0.001$, ****$P < 0.0001$). Data represents mean SD. The data underlying this figure can be found in S1 Data.

correlated with disease severity, with the highest levels observed in myeloid cells from the most severely affected mice (S2A Fig). In contrast, no change in SLAMF5 expression was detected on non-myeloid brain cells (CD45+CD11b−), nor was there any correlation between SLAMF5 levels and disease severity in this population (S2B and S2C Fig). These findings suggest that SLAMF5 upregulation in the disease state is specific to brain myeloid cells.

To further investigate the role of SLAMF5 on myeloid cells, we examined the expression of the activation markers MHCII and CD80, which are indicative of antigen-presenting cell activation [52]. SLAMF5-deficient myeloid cells exhibited significantly reduced expression of both MHCII (Fig 1C and 1D) and CD80 (Fig 1E and 1F) compared to WT controls. Furthermore, a positive correlation was observed between the expression levels of these markers and disease severity (S3A and S3B Fig), suggesting that SLAMF5 contributes to myeloid cell activation in the context of EAE.

To distinguish microglia from other myeloid cell populations in the brain, we utilized the microglia-specific marker TMEM119 [53]. While TMEM119 is a well-established marker for microglia, its expression can be reduced or absent in some microglial cells, particularly under neuroinflammatory conditions [54]. Based on TMEM119 expression, we identified two distinct myeloid populations: TMEM119-positive microglia and TMEM119-negative myeloid cells. The latter group includes TMEM119-negative microglia, BAMs, and infiltrating peripheral macrophages. The gating strategy used to define these populations is shown in S4A Fig. SLAMF5-deficient TMEM119-positive microglia expressed lower levels of MHCII (S4B Fig) and CD80 (S4C Fig) as the total myeloid cells. Thus, both TMEM119-positive and TMEM119-negative cells

were in a less activated state in EAE-induced SLAMF5-deficient mice compared to WT controls, suggesting that the lack of SLAMF5 contributes to the regulation of the immune response and the control of excessive inflammation, ultimately leading to milder disease progression. Since no significant differences were observed between the TMEM119-positive and TMEM119-negative populations, we focused the remainder of the study on the total CD11b-positive myeloid population.

Analysis of the leukocyte population (CD11b⁻ cells) revealed an increased presence of these cells in the brains of sick WT mice. In contrast, SLAMF5-deficient mice exhibited a reduced percentage of CD11b⁻ cells, suggesting that SLAMF5 may play a role in regulating the infiltration and or accumulation of peripheral immune cells, such as T cells, into the brain in the EAE model (Fig 1G).

Our previous study demonstrated a reduced Th1 population in the spinal cord of EAE-induced SLAMF5-deficient mice [36]. Therefore, we next examined the T cell population in the brains of EAE-induced mice. SLAMF5-deficient mice exhibited a reduced number of both CD4⁺ and CD8⁺ T cells, suggesting decreased infiltration, retention, or survival of lymphocytes in the absence of SLAMF5 (Fig 1H and 1I). Furthermore, while levels of IL-17 and FOXP3 in CD4⁺ T cells were comparable between WT and SLAMF5-deficient mice, a significant reduction in interferon-γ (IFN-γ) expression was observed in CD4⁺ T cells from SLAMF5-deficient mice (Fig 1J). This finding suggests a less activated T cell phenotype in the absence of SLAMF5. Interestingly, the altered myeloid cell phenotype and reduced lymphocyte activation occurred concurrently, suggesting a coordinated or compensatory adaptation within the immune response.

To investigate the direct role of SLAMF5 in brain-resident myeloid cells, we generated a conditional SLAMF5 knockout using the *Cx3cr1*^CreER model, which enables tamoxifen (TAM)-inducible gene ablation. TAM treatment of these mice induces recombination of floxed alleles in all *Cx3cr1*⁺ cells [55]. However, over time, the recombined allele is lost in the short-lived monocyte population, whereas in the CNS, only long-lived microglia and BAM retain the genetic modification [56]. The SLAMF5 conditional knockout mice (cKO) were heterozygous for *Cx3cr1*^CreER and homozygous for the floxed *Slamf5* allele, while the control mice were homozygous for the SLAMF5flox gene (Fig 2A).

To validate SLAMF5 deletion, we measured its expression on CD45⁺CX3CR1⁺ cells in the brain. As expected, SLAMF5 expression was significantly reduced in this population in cKO mice (Fig 2B and 2C). In contrast, SLAMF5 expression remained unchanged in brain-derived CD4⁺ T cells and in peripheral myeloid cells (Fig 2D–2G), confirming that the deletion was specific to brain-resident myeloid cells.

The first clinical sign of EAE—distal tail limpness (score=0.5)—was observed on day 11 in the control group, whereas SLAMF5 cKO mice remained below this clinical threshold. By day 15, the mean clinical score in control mice was 2.02±0.234, while cKO SLAMF5 mice exhibited a markedly lower score of 0.125±0.072 (Fig 2H). Furthermore, analysis of the Area Under the Curve (AUC) for clinical scores revealed a significant reduction in disease burden in cKO mice compared to controls (Fig 2I).

Following IACUC regulations, mice reaching a clinical score of 3 or experiencing weight loss exceeding 20% were euthanized, limiting assessment of the full disease course in the control group. However, extended monitoring of cKO mice (Fig 2H) showed that they did not reach these endpoints and exhibited no significant increase in disease score over time. These findings suggest that SLAMF5 expression in brain-resident myeloid cells plays a critical role in both the initiation and progression of EAE.

Consistent with this interpretation, expression levels of MHCII and CD80 on CNS myeloid cells were significantly reduced in cKO mice induced with EAE, while no significant differences were observed in naïve (non-immunized) mice (Fig 2J–2M). Furthermore, SLAMF5 deficiency in brain myeloid cells showed no effect on the expression of MHCII and CD80 on splenic myeloid cells (S5A and S5B Fig).

Together, these results underscore the importance of SLAMF5 in regulating myeloid cell activation within the CNS and its contribution to neuroinflammatory disease progression.

Histopathological analysis of spinal cords from EAE-induced SLAMF5 cKO and control mice revealed significantly reduced lymphocytic infiltration in cKO animals in the cervical, thoracic, and lumbosacral regions, supporting a protective

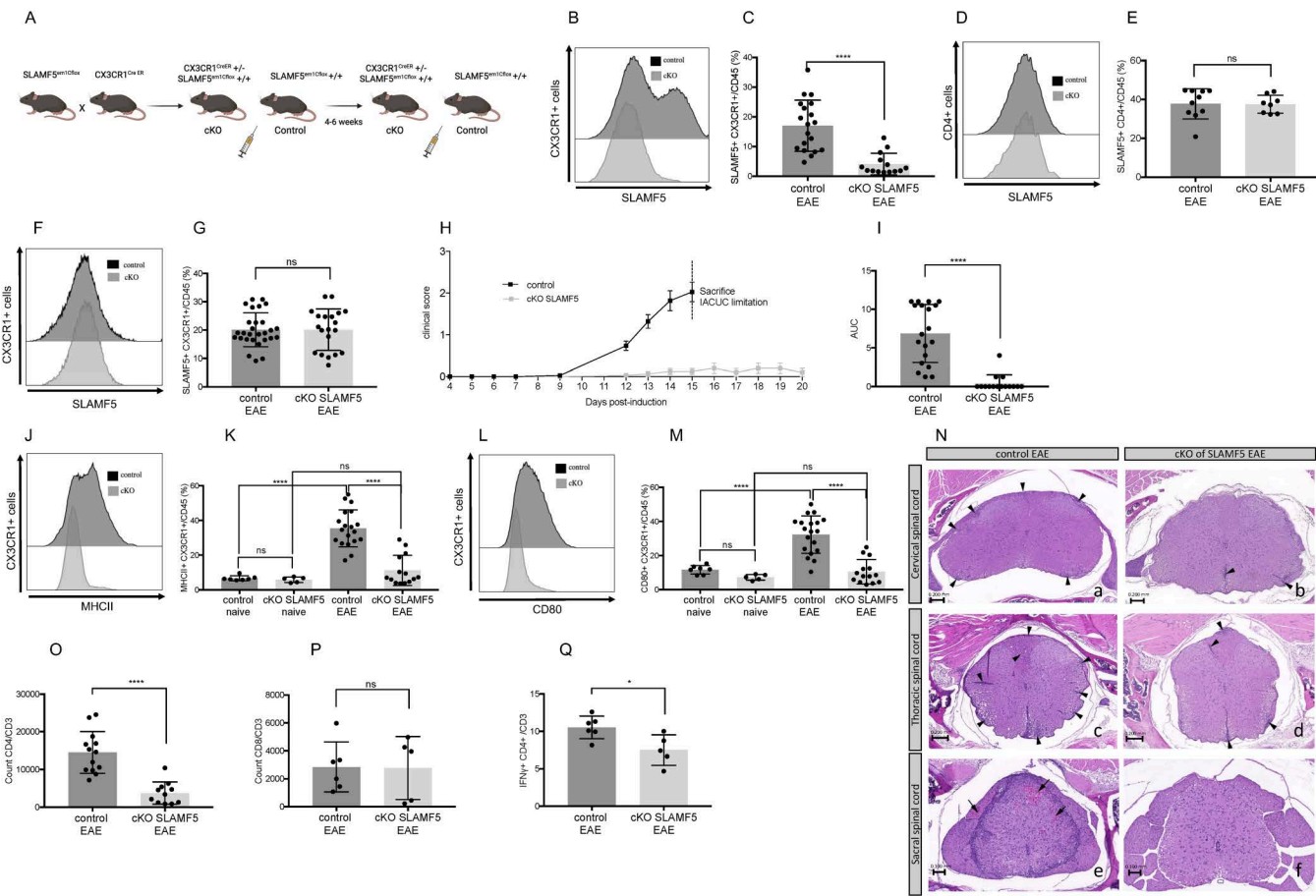

**Fig 2. Conditional knockout of SLAMF5 in CX3CR1er cells attenuates EAE progression, reduces brain myeloid cell activation, and decreases T cell infiltration.** EAE (MOG35-55) was induced in conditional knock out (cKO) of SLAMF5 mice and control mice. Mice were monitored for 15 days. On day 15, the mice were sacrificed, and their brains and spleens were excised. Brain and spleen immune cells were isolated, stained, and analyzed by FACS. Dead cells were excluded from analysis by Zombie Live/Dead staining. **(A)** Schematic of mouse genotypes and experimental approach. **(B, C)** Analysis of SLAMF5 expression in brain myeloid cells (CD45+ CX3CR1+) of control and cKO EAE-induced mice. **(B)** Representative histograms, and **(C)** bar chart shows SLAMF5 expression in % in brain myeloid population (control $n = 19$, cKO of SLAMF5 $n = 14$). **(D, E)** Analysis of SLAMF5 expression in T cells in the brain (CD3+ CD4+) (control $n = 10$, cKO of SLAMF5 $n = 8$). **(F, G)** Analysis of SLAMF5 expression in splenic myeloid cells (CD45+ CX3CR1+) (control $n = 28$, cKO of SLAMF5 $n = 20$). Graphs show five independent repeats. Unpaired Student $t$ test with 95% confidence levels. **(H)** Daily mean clinical scoring of the disease. **(I)** Bar graph shows the Area Under the Curve of the clinical score. Mann-Whitney test. Data represent mean ± SEM. **(J, K)** Analysis of MHCII expression in brain myeloid cells (CD45+ CX3CR1+). **(J)** Representative histograms, and **(K)** Bar chart shows SLAMF5 expression in % in brain myeloid population. **(L, M)** Analysis of CD80 expression in brain myeloid cells (CD45+ CX3CR1+) (control naïve $n = 8$, cKO of SLAMF5 naïve = 5, control EAE $n = 19$, cKO of SLAMF5 EAE $n = 14$). Graphs show three independent repeats. Ordinary one-way ANOVA with Dunnett multiple comparison tests. **(N)** Representative samples from the cervical (a + b), thoracic (c + d), and sacral (e + f) spinal cord comparing the degree of inflammation and white matter damage in control (a, c, e) and cKO (b, d, f) mice. Arrowheads mark foci with lymphocytic-dominant infiltration. In some cases, the infiltrate is associated with white matter damage, seen as pallor at this magnification. **(O)** Count of CD3 and CD4 positive cells in the brain (control EAE $n = 13$; cKO of SLAMF5 $n = 11$). Graph shows four independent determinations. **(P)** Count of CD3 and CD8 positive cells in the brain. **(Q)** Expression of Interferon γ in % in CD3 CD4 positive T cells (control EAE, $n = 6$; cKO of SLAMF5, $n = 5$). Graph shows two independent determinations. Two-tailed unpaired Student $t$ test with 95% confidence levels (*$P < 0.05$, **$P < 0.01$, ***$P < 0.001$, ****$P < 0.0001$). Data represents mean SD. The data underlying this figure can be found in S1 Data.

role for SLAMF5 deficiency in brain-resident myeloid cells. In the cervical and thoracic regions, arrowheads indicate inflammatory foci, while in the sacral region of control samples inflammation encircles the cord and arrows indicate multi-focal hemorrhage (Fig 2N).

In the brain, flow cytometric analysis showed a reduced number of CD4⁺ T cells in cKO mice, with no significant change in CD8⁺ T cell infiltration (Fig 2O and 2P). Additionally, IFN-γ production by CD4⁺ T cells was significantly decreased in cKO mice, further indicating diminished T cell activation in the absence of SLAMF5 signaling in CNS myeloid cells (Fig 2Q).

These findings demonstrate that SLAMF5 expression in brain-resident myeloid cells contributes to CNS inflammation during EAE by promoting both the infiltration and activation of CD4⁺ T cells. The observed reduction in lymphocyte presence within the spinal cord and brain, along with decreased IFN-γ production by CD4⁺ T cells in cKO mice, highlights a key immunoregulatory role for myeloid cell–derived SLAMF5 in driving neuroinflammatory responses.

## Direct SLAMF5 inhibition in the CNS significantly attenuates disease progression

To further investigate the in vivo role of SLAMF5 in EAE, we administered a SLAMF5-blocking antibody or an IgG2a isotype control to EAE-induced mice via intravenous (I.V.) injection on days 7, 9, and 12 postinduction, following the protocol previously described by Radomir and colleagues [36]. Mice were monitored daily for clinical signs of EAE for up to 15 days using a standard scoring system. Consistent with previous findings, SLAMF5 blockade had a partial effect on disease onset and reduced overall disease severity compared to control-treated mice (S6A and S6B Fig).

We next examined the impact of SLAMF5 blockade on CNS myeloid cell activation by assessing the expression levels of MHCII and CD80 on CD11b⁺ myeloid cells. Treatment with the SLAMF5-blocking antibody did not alter the expression of either marker (Fig 3A and 3B), indicating that SLAMF5 inhibition does not directly affect the activation status of CNS-resident myeloid cells under these conditions. These findings suggest that the observed reduction in disease severity following systemic SLAMF5 blockade is likely mediated through modulation of peripheral immune responses, particularly regulatory B cells, as previously reported [36].

Given that SLAMF5 deficiency regulates myeloid cell activation, we hypothesized that the lack of an effect on CNS myeloid cell activation in IV antibody-treated mice may be due either to the resistance of brain-resident myeloid cells to antibody-mediated inhibition or to limited antibody access across the BBB. To address this point, we investigated whether SLAMF5 blockade directly regulates microglial activation in vitro. The N9 microglial cell line was stimulated with LPS, as previously described [57], to induce M1-like polarization, thereby mimicking key pro-inflammatory features observed in neuroinflammatory conditions such as EAE. Following LPS stimulation, N9 microglial cells exhibited a significant upregulation of MHCII (~1.75-fold) and CD80 (~8.5-fold), consistent with the activation phenotype of microglia and antigen-presenting cells observed in vivo during EAE induction. When SLAMF5 was blocked during the final 24 h of LPS stimulation, MHCII expression was reduced to levels comparable to those seen in unstimulated cells. Similarly, CD80 expression was significantly decreased in the presence of the SLAMF5-blocking antibody (Fig 3C–3F). These results indicate that SLAMF5 contributes to microglial activation under inflammatory conditions, and that its inhibition can dampen this response.

To further assess the direct effect of SLAMF5 on myeloid cells, we isolated primary myeloid cells from EAE-induced WT mice and incubated them with either a SLAMF5-blocking antibody or an isotype control. The expression of MHCII and CD80 was then analyzed. As shown in Fig 3G–3J, SLAMF5 blockade led to a marked downregulation of both MHCII and CD80 expression, indicating that SLAMF5 directly contributes to the activation of myeloid cells.

Together, these findings support the conclusion that SLAMF5 plays a regulatory role in promoting myeloid cell activation within a pro-inflammatory environment.

Given that the SLAMF5-blocking antibody modulates brain myeloid cell activation in vitro but not in vivo via systemic administration, we next sought to evaluate its effects on disease progression and CNS myeloid activation when delivered directly into the CNS. To bypass the BBB, SLAMF5-blocking or isotype control antibodies were administered into the cerebrospinal fluid via ICV injection on day 5 following EAE induction in WT mice (Fig 3K).

ICV administration of the SLAMF5-blocking antibody significantly prevented disease development compared to isotype control-treated mice. By day 15, the mean clinical score of the control group was 2.3 ± 0.25, whereas the SLAMF5-blocked group exhibited a markedly lower score of 0.3 ± 0.1 (Fig 3L and 3M). As expected, SLAMF5 expression was

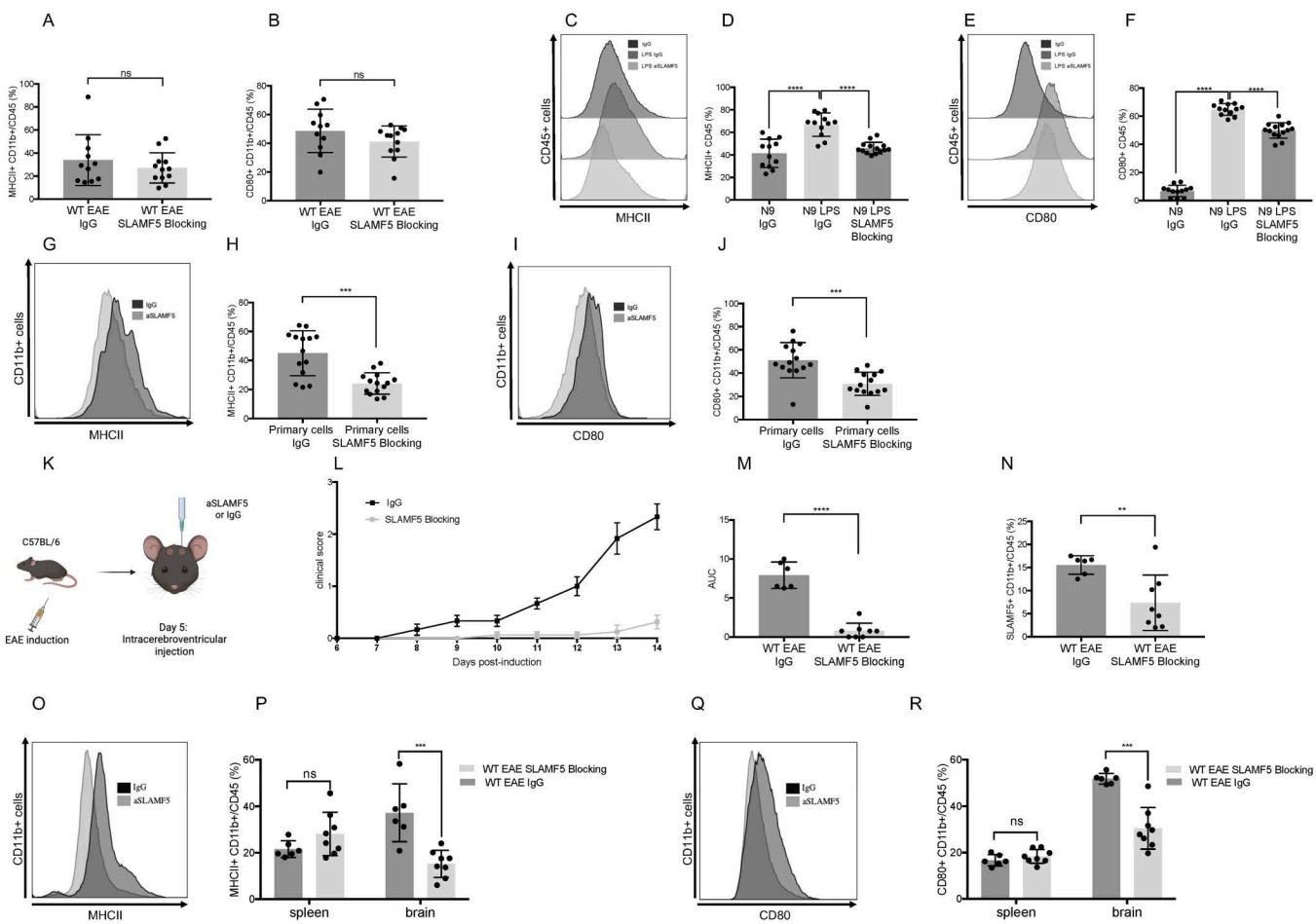

**Fig 3. Blocking SLAMF5 in myeloid cells in vitro and via intracerebroventricular injection in vivo reduces their activation and attenuates neuroinflammation. (A, B)** EAE (MOG35-55) was induced in WT mice. Mice were treated intravenously with three injections of either 150 μg of SLAMF5 blocking antibody or an IgG control on days 7, 9, and 12 postinduction. On day 15, the mice were sacrificed, and their brains were excised. The brains were processed, and immune cells were isolated, stained, and analyzed by FACS. Dead cells were excluded from analysis by Zombie Live/Dead staining. Bar graphs showing: **(A)** The expression of MHCII in % in CD45+ CD11b+ brain myeloid cells. **(B)** Expression of CD80 in % in CD45+ CD11b+ brain myeloid cells (SLAMF5-blocking n = 12; IgG *n* = 11). The bar graphs show three independent repeats. Two-tailed unpaired Student *t* test with 95% confidence levels **(C–F)** N9 cells were seeded and treated with 10 μg LPS for four consecutive days; during the last 24 h, the LPS-induced N9 cells were treated with 60 μg of the SLAMF5 blocking antibody or the IgG control. The cells were stained and analyzed by FACS. Dead cells were excluded from analysis by Zombie Live/Dead staining. Representative histogram and Bar graph show: (C, D) The expression of MHCII in the CD45+ cells. (E, F) the expression of CD80 in the CD45+ cells (IgG *n* = 12, LPS IgG *n* = 12; LPS SLAMF5 Blocking *n* = 14). Graphs show three independent repeats. Ordinary one-way ANOVA with Dunnett multiple comparison tests. **(G–J)** EAE (MOG35-55) was induced in WT mice. At the peak of the disease, the mice were sacrificed, and their brains were excised, and immune cells were isolated. CD11b positive cells were separated by microbeads, and the CD11b positive cells were incubated with 60 μg of the SLAMF5 blocking antibody or with the IgG control antibody for 48 h. The cells were stained and analyzed by FACS. Dead cells were excluded from analysis by Zombie Live/Dead staining. Representative histogram and bar graphs show: (G, H) The expression of MHCII in primary CD45+ CD11b+ brain myeloid cells. (I, J) The expression of CD80 in primary brain CD45+ CD11b+ myeloid cells (SLAMF5-blocking *n* = 14; IgG *n* = 14). Graphs show three independent repeats. Two-tailed unpaired Student *t* test with 95% confidence levels. **(K–R)** EAE (MOG35-55) was induced in WT mice and treated by Intracerebroventricular injection (ICV) at day 5 postinduction with 3 μg/ventricle of the SLAMF5 blocking antibody or the IgG control. On day 14, the mice were sacrificed, and their brains and spleens were excised. The brains and spleens were processed, and immune cells were isolated, stained, and analyzed by FACS. Dead cells were excluded from analysis by Zombie Live/Dead staining. (K) Schematic of the experimental approach. (L) Graph shows the daily mean clinical scoring of the disease. (M) Bar graph shows the Area Under the Curve of the clinical score. Mann–Whitney test. Data represent mean ± SEM. Bar graph shows: (N) The expression of SLAMF5 in % in CD45+ CD11b+ brain myeloid cells. Representative histogram and bar graphs show: (O, P) The expression of MHCII in CD45+ CD11b+ brain and splenic myeloid cells. (Q, R) The expression of CD80 in CD45+ CD11b+ brain and splenic myeloid cells. (SLAMF5-blocking *n* = 8; IgG *n* = 6). Graphs show three independent experiments. Ordinary one-way ANOVA with Dunnett multiple comparison tests. (*P < 0.05, **P < 0.01, ***P < 0.001, ****P < 0.0001). Data represents mean SD. The data underlying this figure can be found in S1 Data.

downregulated in brain myeloid cells following ICV injection of the blocking antibody (Fig 3N). Importantly, this treatment also led to a significant reduction in the expression of activation markers MHCII and CD80 on brain CD45+CD11b+ myeloid cells, while no such changes were observed in splenic myeloid cells (Fig 3O–3R).

These findings confirm that SLAMF5 blockade within the CNS can effectively suppress myeloid cell activation and attenuate disease initiation and progression in EAE, reinforcing the importance of CNS-specific SLAMF5 signaling in neuroinflammation.

## SLAMF5 is a positive regulator of CD52 expression in myeloid cells through BHLEH40

To further investigate the downstream signaling cascades regulated by SLAMF5 in myeloid cells, we performed RNA sequencing (RNA-seq) on purified mRNA from sorted CNS myeloid cells isolated from EAE-induced WT and SLAMF5−/− (KO) mice.

Principal Component Analysis (PCA) revealed distinct clustering patterns associated with disease severity. WT mice with high clinical scores (score 3) formed a separate cluster from those with milder symptoms (scores 0.5–1), reflecting transcriptional changes linked to disease progression. In contrast, SLAMF5 KO mice—who exhibited relatively uniform disease severity—clustered tightly together, indicating consistent transcriptional profiles within this group (Fig 4A).

Despite variability in disease scores among WT animals, RNA-seq identified significant changes in gene expression in the SLAMF5 KO samples. A total of 453 genes were differentially expressed (absolute fold change > 2; $p < 0.05$), with 395 genes downregulated and 58 upregulated in KO mice compared to WT controls (Fig 4B). Expression of these genes also correlated with disease severity: the most severely affected WT mice (score = 3) showed higher expression of upregulated genes compared to WT mice with milder disease (scores = 1 and 0.5) (Fig 4C). These findings underscore SLAMF5's broad regulatory role in EAE pathogenesis.

Pathway enrichment analysis of SLAMF5-regulated genes revealed strong associations with immune-related and neuroinflammatory pathways, Th1 and Th2 differentiation, interferon signaling, agranulocyte adhesion and diapedesis, IRF activation, neuroinflammation signaling, iNOS signaling, and pathways implicated in the pathogenesis of MS. Furthermore, in SLAMF5-deficient mice, multiple genes involved in antigen presentation—such as MHCII molecules and co-stimulatory markers—were significantly downregulated, suggesting that SLAMF5 positively regulates the antigen presentation capacity of myeloid cells (Fig 4D).

Notably, SLAMF5 regulated the expression of several key genes previously associated with myeloid cell activation and EAE pathogenesis, including *Cd52, Irf7, Irf1, Ddx58, Itgax, Vcam1, Stat1, Bhlhe40,* and *Hspa1a* (Fig 4E). RT-qPCR analysis validated the RNA-seq findings, confirming that these genes were significantly downregulated in SLAMF5−/− myeloid cells from EAE-induced mice. Moreover, their expression was elevated in myeloid cells from EAE-induced WT mice compared to healthy controls, confirming their induction in neuroinflammatory conditions. Interestingly, *Hspa1a* was upregulated in SLAMF5−/− cells, consistent with its previously reported neuroprotective role in myeloid cells (S7 Fig) [58].

To further validate these findings in a model of conditional SLAMF5 deletion, we performed RT-qPCR on sorted CNS myeloid cells from EAE-induced SLAMF5 cKO mice. As shown in Fig 5A, the same set of SLAMF5-regulated genes identified by RNA-seq were also downregulated in cKO myeloid cells, indicating that SLAMF5 modulates a consistent gene expression program in both full and cKO models. In addition, we evaluated SLAMF5's role in regulating gene expression in vitro using the N9 microglial cell line. Cells were stimulated with LPS and treated with either a SLAMF5-blocking antibody or an isotype control. In LPS with control antibody-treated cells, we observed strong upregulation of pro-inflammatory genes including *Irf7, Ddx58, Itgax, Vcam1, Irf1, Stat1,* and *Bhlhe40*. In contrast, SLAMF5 blockade significantly reduced the expression of these genes (Fig 5B), further supporting SLAMF5's role in regulating myeloid activation under inflammatory conditions.

One of the most significantly downregulated genes upon SLAMF5 blockade was *CD52* (Figs 4E and S7). CD52 encodes a glycosylphosphatidylinositol (GPI)-anchored cell surface glycoprotein expressed on various leukocyte

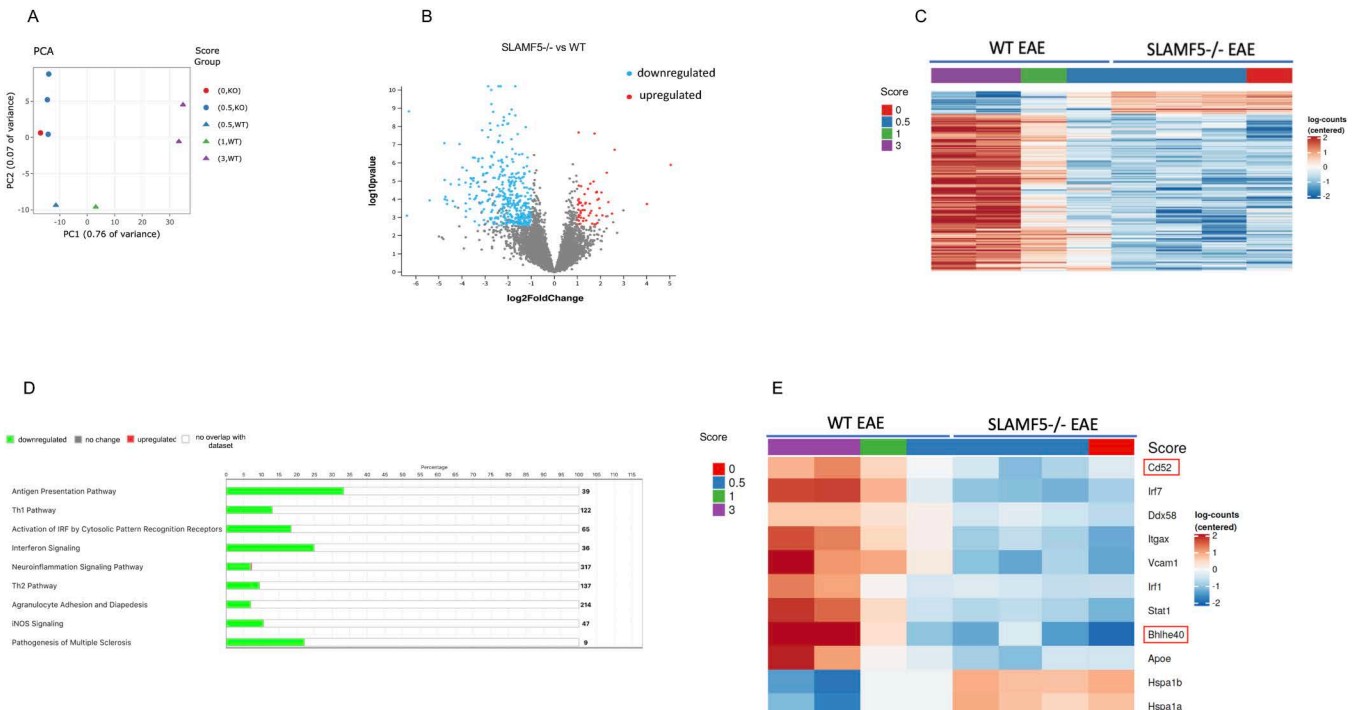

**Fig 4. SLAMF5 acts as a positive regulator of neurodegeneration, demyelination, and leukocyte migration in EAE.** EAE (MOG35-55) was induced in WT and in SLAMF5-deficient mice. At the peak of the disease, the mice were sacrificed, and their brains were excised. Their brains were processed and immune cells were isolated, and stained for dead cells, and for CD45 and CD11b. The double positive population was sorted BD FACS Melody. RNA was extracted from the cells and analyzed (EAE-induced WT group $n = 4$; EAE-induced SLAMF5−/− group $n = 4$). **(A)** Principal component analysis (PCA) of 1,000 most variable genes in brain myeloid cells in WT and SLAMF5−/− induced mice and clustered by clinical score. **(B)** Volcano plot depicting differentially expressed genes in SLAMF5−/− compared to WT. Right dots represent genes expressed at higher levels in SLAMF5−/− myeloid cells, while left dots represent genes with higher expression levels in WT myeloid cells. Y-axis denotes −log10 P values, while X-axis shows log2 fold change values. Red and Blue dots are the differentially expressed genes (absolute change > 2; P value < 0.05). **(C)** Heatmap shows differentially expressed genes grouped by clinical score of EAE-induced mice. The colored bar indicates the standardized log2 normalized counts. **(D)** Pathway enrichment analysis of 453 genes, with 395 genes downregulated and 58 genes upregulated in SLAMF5−/− (absolute change > 2; P value < 0.05). The analysis was done using the IPA QIAGEN platform. The graph shows the number of genes that are downregulated (green), or upregulated (red) in each pathway. The percentage represents the ratio between the number of molecules detected in the sequencing vs. the total number of molecules that are involved in each pathway. **(E)** Heatmap of a total of 11 genes that were differentially expressed between WT EAE and SLAMF5−/− EAE mice. The colored bar indicates the standardized log2 normalized counts. These genes were shown to be involved in EAE and in myeloid cell functions. The data underlying this figure can be found in S1 Data.

populations, including lymphocytes, monocytes, and myeloid cells. While the function of CD52 in myeloid cells remains largely undefined [37], its clinical relevance in MS is well established. The CD52-targeting monoclonal antibody alemtu-zumab is approved for the treatment of adult MS patients by both the European Medicines Agency and the U.S. Food and Drug Administration [59,60].

Downregulation of CD52 protein expression was observed in total CD45⁺/CD11b⁺ myeloid cells isolated from SLAMF5-deficient mice (Fig 5C and 5D) as well as in brain-resident myeloid cells from SLAMF5 cKO mice (Fig 5E and 5F). Furthermore, myeloid cells from EAE-induced mice that received ICV injections of the SLAMF5-blocking antibody exhibited reduced CD52 expression in brain-derived myeloid populations (Fig 5G and 5H). These findings suggest that SLAMF5 positively regulates CD52 expression in myeloid cells during disease progression.

To further validate the role of SLAMF5 in regulating CD52 expression, we performed an in vitro assay using primary myeloid cells isolated from the brains of EAE-induced WT mice. Cells were cultured for 48 h in the presence of either

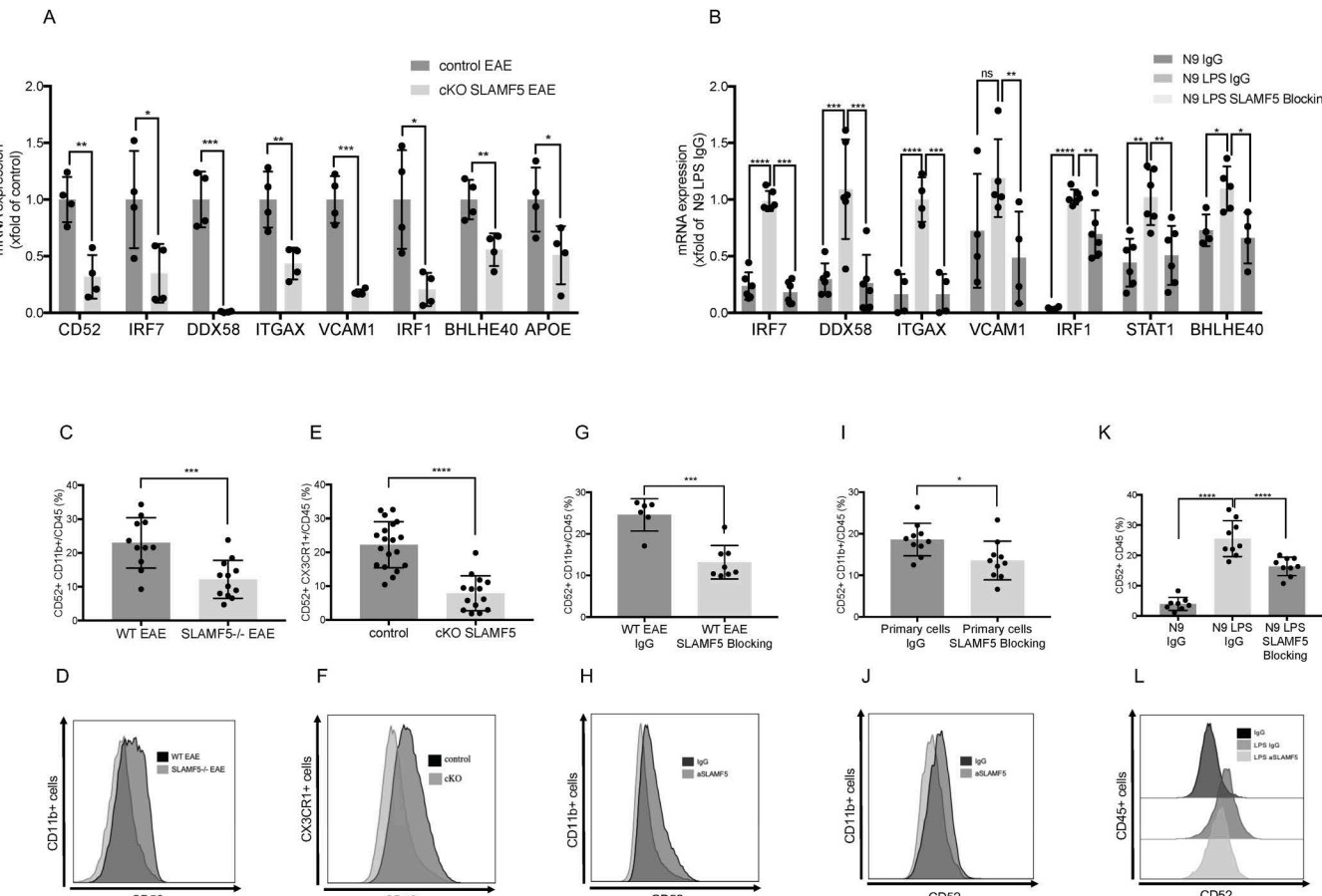

**Fig 5. SLAMF5 regulates myeloid cells activation in a CD52-dependent manner. (A)** EAE (MOG35-55) was induced in control and in SLAMF5 cKO mice. At the peak of the disease, the mice were sacrificed, and their brains were excised. The brains were processed, and immune cells were isolated, stained for dead cells, and labeled with anti-CD45 and anti-CD11b antibodies. The double-positive population was sorted. RNA was isolated and cDNA was synthesized. The expression levels of genes were determined by qRT-PCR. Data are presented as fold change relative to the control group. Two-tailed unpaired Student $t$ test with 95% confidence levels. Graphs show two independent determinations. **(B)** N9 cells were seeded and treated with 10 μg LPS for two consecutive days; during the last 24 h the LPS-induced N9 cells were treated with 60 μg of the SLAMF5 blocking antibody or the IgG control. RNA was isolated and cDNA was synthesized. The expression levels of genes were determined by qRT-PCR. Data are presented as fold change relative to the control group N9 treated with LPS and IgG control. Ordinary one-way ANOVA with Dunnett multiple comparison tests. Graphs show two independent determinations. **(C, D)** EAE (MOG35-55) was induced in WT and in SLAMF5-deficient mice. Mice were sacrificed at the peak of the disease. Dead cells were excluded from analysis by Zombie Live/Dead staining. Analysis of CD52 expression in brain myeloid cells (CD45+ CD11b+) (EAE-induced WT $n=11$; EAE-induced SLAMF5−/− $n=12$). Graphs show three independent repeats. Two-tailed unpaired Student $t$ test with 95% confidence levels. **(E, F)** EAE (MOG35-55) was induced in cKO SLAMF5 mice and control mice. At the peak of the disease, the mice were sacrificed, and their brains were excised. Their brains were processed, and immune cells were isolated, stained, and analyzed by FACS. Dead cells were excluded from analysis by Zombie Live/Dead staining. Analysis of CD52 expression in brain myeloid cells (CD45+ CX3CR1+) (control=19; cKO of SLAMF5 n=14). Graphs show five independent repeats. Two-tailed unpaired Student $t$ test with 95% confidence levels. **(G, H)** EAE (MOG35-55) was induced in WT mice, and the mice were treated by Intracerebroventricular injection at day 5 postinduction with 3 μg/ventricle of the SLAMF5 blocking antibody or IgG control. At the peak of the disease, the mice were sacrificed, and their brains were excised. The brains were processed, and immune cells were isolated, stained, and analyzed by FACS. Dead cells were excluded from analysis by Zombie Live/Dead staining. Analysis of CD52 expression in brain myeloid cells (CD45+ CD11b+). (SLAMF5-blocking $n=8$; IgG $n=6$). Graphs show three independent repeats. Two-tailed unpaired Student $t$ test with 95% confidence levels. **(I, J)** EAE was induced in WT mice. At the peak of the disease, the mice were sacrificed, and their brains were excised. Their brains were processed, and immune cells were isolated. CD11b positive cells were separated by microbeads, and the positive cells were incubated with 60 μg of the SLAMF5 blocking antibody or with the IgG control antibody for 48 **h.** The cells were stained and analyzed by FACS. Dead cells were excluded from analysis by Zombie Live/Dead staining. Analysis of CD52 expression in primary brain myeloid cells CD45+ CD11b+ (SLAMF5-blocking $n=10$; IgG $n=10$). Graphs show three independent repeats. Two-tailed unpaired Student $t$ test with 95% confidence levels. **(K, L)** N9 were seeded and treated one time with 10 μg LPS for four consecutive days; for the last 24 h. The LPS-induced N9 cells were treated with 60 μg of the SLAMF5 blocking or the IgG control antibodies. The cells were stained and analyzed by FACS. Dead cells were excluded from analysis by Zombie Live/Dead staining. Analysis of CD52 expression in CD45+ cells (IgG $n=8$, LPS IgG $n=9$; LPS SLAMF5 Blocking $n=9$). Graphs show three independent repeats. Ordinary one-way ANOVA with Dunnett multiple comparison tests. Graphs show two independent determinations. (*$P<0.05$, **$P<0.01$, ***$P<0.001$, ****$P<0.0001$). Data represents mean SD. The data underlying this figure can be found in S1 Data.

the SLAMF5-blocking antibody or an isotype control. As shown in Fig 5I and 5J, SLAMF5 blockade significantly reduced CD52 expression in these primary myeloid cells.

Additionally, analysis of the N9 microglial cell line demonstrated that LPS stimulation increased CD52 expression compared to non-stimulated cells. However, treatment with the SLAMF5-blocking antibody significantly downregulated CD52 expression in LPS-activated N9 cells relative to the isotype control (Fig 5K and 5L).

Together, these results provide compelling evidence that SLAMF5 regulates CD52 expression in CNS myeloid cells both in vivo and in vitro, particularly under inflammatory conditions associated with neuroinflammation and EAE progression.

To investigate the role of CD52 in myeloid cell activation, we downregulated CD52 expression in N9 microglial cells using siRNA (S8A Fig). Following CD52 knockdown, cells were stimulated with LPS, and the expression of activation markers was assessed. While no significant changes were observed in non-activated cells, CD52 knockdown led to a marked reduction in MHCII (Fig 6A and 6B) and CD80 (Fig 6C and 6D) expression in LPS-activated N9 cells. These results suggest that CD52 contributes to myeloid cell activation under inflammatory conditions. Taken together with earlier data, these findings support the conclusion that SLAMF5 regulates myeloid cell activation, at least in part, through modulation of CD52 expression.

Another gene identified as SLAMF5-regulated in our RNA-seq dataset was *BHLHE40*, a transcription factor previously implicated in neuroinflammation and disease progression in the EAE model [61]. Expression of BHLHE40 was upregulated in CNS myeloid cells from WT mice and significantly downregulated in SLAMF5-deficient mice (Figs 4E and S7). To evaluate its functional role, we downregulated BHLHE40 expression in N9 cells using siRNA (S8B Fig), followed by LPS stimulation. MHCII (Fig 6E and 6F), CD80 (Fig 6G and 6H), and CD52 (Fig 6I and 6J) expression levels were all significantly reduced in BHLHE40-silenced LPS-activated cells. In contrast, BHLHE40 knockdown had no effect on these markers in unstimulated cells.

These results indicate that BHLHE40 is a critical regulator of myeloid cell activation and CD52 expression under pro-inflammatory conditions and suggest that SLAMF5 exerts its effects on myeloid activation through a BHLHE40-dependent pathway.

To determine whether SLAMF5 regulates *CD52* expression via modulation of *BHLHE40* transcriptional activity, we assessed the binding of BHLHE40 to the CD52 promoter using chromatin immunoprecipitation followed by quantitative RT-PCR (ChIP-qRT-PCR) as was previously described [48,49]. Brain-derived CD45+CD11b+ myeloid cells were isolated from EAE-induced mice and incubated *ex vivo* with either a SLAMF5-blocking antibody or an isotype control. ChIP-qRT-PCR analysis revealed a significant reduction in BHLHE40 binding to the *CD52* promoter in cells treated with the SLAMF5-blocking antibody (Fig 6K). To validate the specificity of the ChIP results, we included *IL10RA*, a known BHLHE40 binding target [62], as a positive control, and *TMEM119*, a gene not reported to bind BHLHE40, as a negative control. The observed binding patterns confirmed the specificity of the assay (S9 Fig). Therefore, we can suggest that SLAMF5 signaling promotes *CD52* expression by regulating *BHLHE40* expression and its recruitment to the *CD52* promoter, thereby contributing to myeloid cell activation.

## SLAMF5 is a positive regulator of myeloid cells in the periphery of EAE mice and in human samples

To explore the therapeutic potential of SLAMF5 as a target in human disease, we first assessed whether SLAMF5 regulates peripheral myeloid cell activation in mice. Splenocytes from WT and SLAMF5-deficient EAE-induced mice were analyzed for the expression of MHCII (Fig 7A), CD80 (Fig 7B), and CD52 (Fig 7C). Lower expression levels of all three markers were observed on CD11b+/F4/80+ splenic myeloid cells from SLAMF5-deficient mice, indicating that SLAMF5 promotes peripheral myeloid cell activation during EAE, while in the cKO, the level of MHCII, CD80 (S5 Fig) and CD52 (S10 Fig) were unchanged. Moreover, the severity of EAE correlated with CD80 expression on splenic myeloid cells (S11 Fig), reinforcing the functional significance of this pathway.

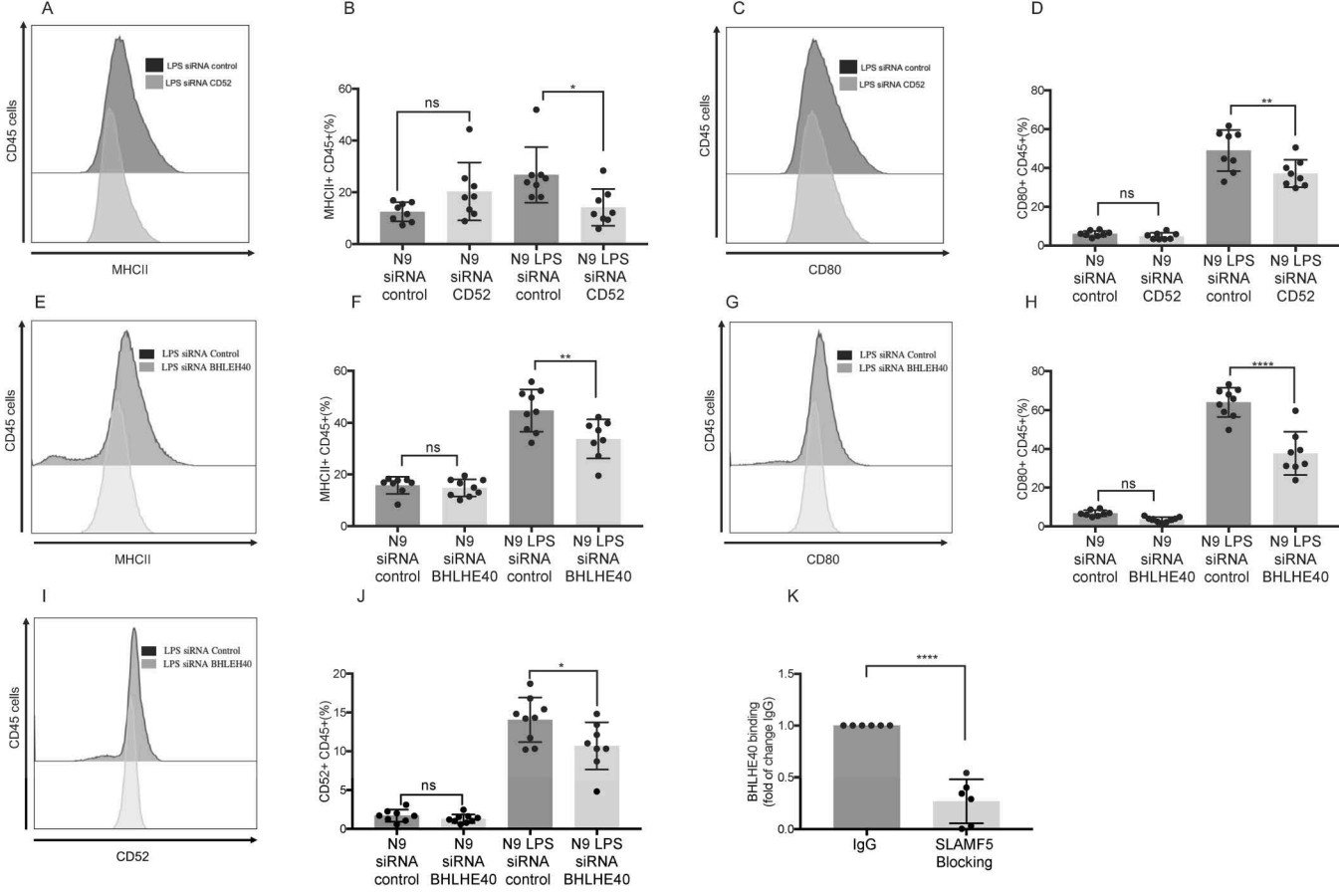

**Fig 6. SLAMF5 regulates CD52 expression in myeloid cells through BHLHE40 binding. (A–D)** N9 were electroporated with CD52 siRNA and Control siRNA. The cells were activated with LPS to mimic a proinflammatory environment for 48h. The cells were stained and analyzed by FACS. Dead cells were excluded from analysis by Zombie Live/Dead staining. Representative and bar graphs show: (A, B) The expression of MHCII on the CD45+ cells. (C, D) The expression of CD80 on the CD45+ cells (LPS-induced N9 with CD52 siRNA, $n=8$; LPS-induced N9 with control siRNA=8). Graphs show three independent repeats. Ordinary one-way ANOVA with Dunnett multiple comparison tests. Graphs show two independent determinations. **(E–J)** N9 cells were electroporated with BHLEH40 siRNA or Control siRNA. The cells were activated with LPS to mimic a pro-inflammatory environment for 48h. The cells were stained and analyzed by FACS. Dead cells were excluded from analysis by Zombie Live/Dead staining. Representative and bar graphs show: (E, F) The expression of MHCII on the CD45+ cell population. (G, H) The expression of CD80 on the CD45+ cells. (I, J) The expression of CD52 on the CD45+ cells (LPS-induced N9 with BHLHE40 siRNA $n=9$; LPS-induced N9 with control siRNA, $n=8$). Graphs show three independent determinations. Ordinary one-way ANOVA with Dunnett multiple comparison tests. Graphs show two independent determinations. **(K)** EAE (MOG35-55) was induced in WT mice. At the peak of the disease, the mice were sacrificed, and their brains were excised. The brains were processed, and immune cells were isolated, stained for dead cells, and labeled with anti-CD45 and anti-CD11b antibodies. The double positive population was sorted by ARIA III 5 laser. The sorted cells were incubated with SLAMF5 blocking antibody or with the isotype control for 1 h. Then, cells were harvested, and ChIP was performed. Binding of BHLHE40 to the promoter area of the CD52 gene was determined by RT-qPCR. The graph presents the percent enrichment of the input. Data are presented as fold change relative to the control group IgG which was normalized to a value of 1. (SLAMF5-blocking $n=6$; IgG $n=6$). Graphs show two independent repeats. Two-tailed unpaired Student $t$ test with 95% confidence levels. ($*P<0.05$, $**P<0.01$, $***P<0.001$, $****P<0.0001$). Data represents mean SD. The data underlying this figure can be found in S1 Data.

To further validate these findings, splenocytes from EAE-induced WT mice were incubated *ex vivo* with either SLAMF5-blocking or isotype control antibodies. Treatment with the blocking antibody resulted in reduced expression of MHCII, CD80, and CD52 on myeloid cells (Fig 7D–7F), further confirming that SLAMF5 modulates activation of peripheral myeloid cells.

Given these findings in murine models, we next investigated whether SLAMF5 plays a similar regulatory role in human monocytes. Peripheral blood leukocytes from newly diagnosed, untreated MS patients and healthy controls were

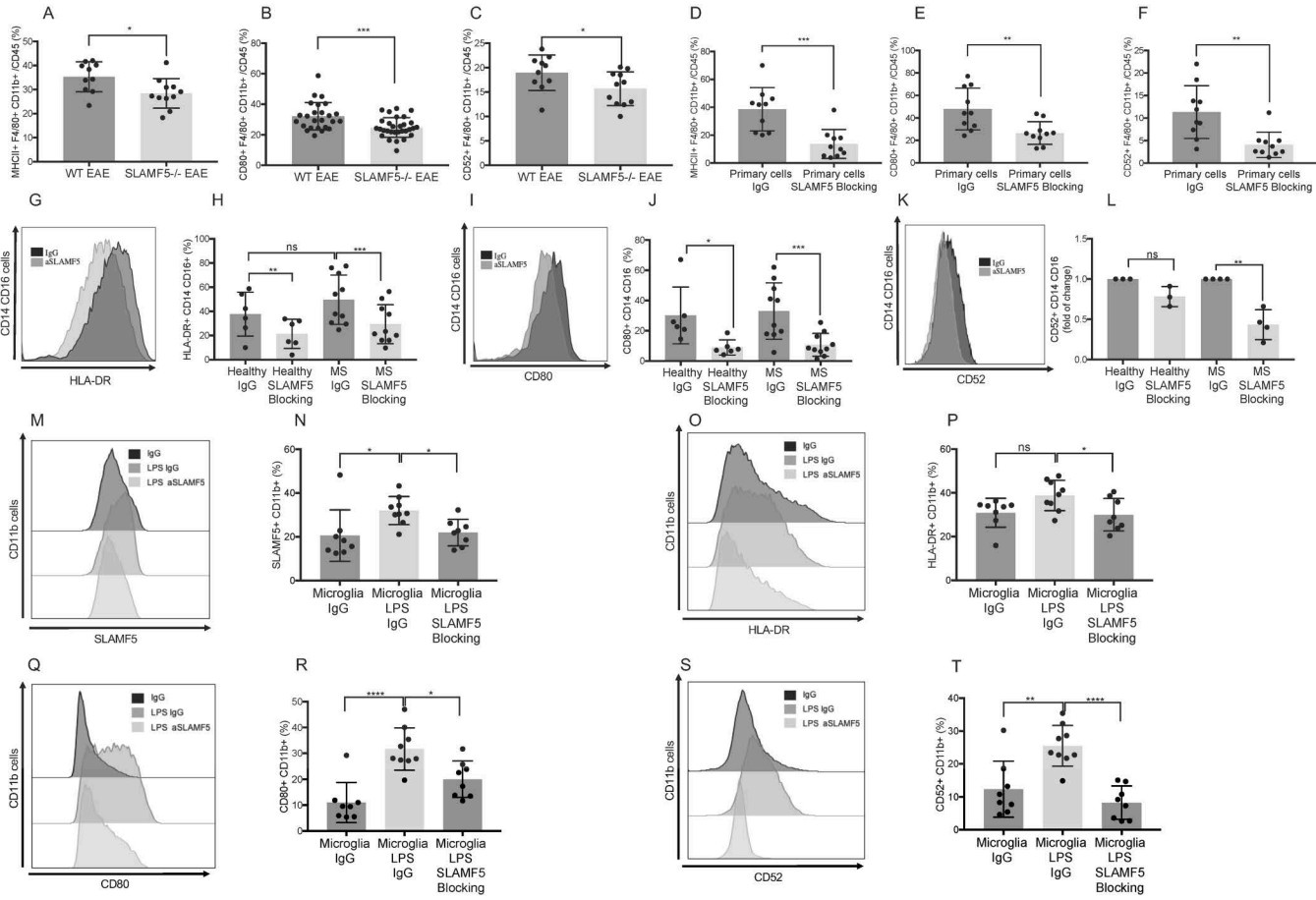

**Fig 7. SLAMF5 modulates peripheral myeloid cell activation in both EAE mice and human samples. (A–C)** EAE (MOG35-55) was induced in WT and SLAMF5-deficient mice. Mice were monitored for 15 days. On day 15, the mice were sacrificed, and their spleens were excised. The spleens were processed, and immune cells were isolated, stained and analyzed by FACS. Dead cells were excluded from analysis by Zombie Live/Dead staining. Bar graphs show: (A) The expression of MHCII in % in CD45+ CD11b+ F4/80+ myeloid cells (WT $n = 10$; SLAMF5−/− $n = 11$). (B) The expression of CD80 in % in CD45+ CD11b+ F4/80+ myeloid cells (WT $n = 25$; SLAMF5−/− $n = 30$). (C) The expression of CD52 in % in CD45+ CD11b+ F4/80+ myeloid cells (WT $n = 10$; SLAMF5−/− $n = 11$). Graphs show three independent repeats. Two-tailed unpaired Student $t$ test with 95% confidence levels. **(D–F)** EAE (MOG35s55) was induced in WT mice. At the peak of the disease, the mice were sacrificed, and their spleens were excised. Spleens were processed, and immune cells were isolated. CD11b positive cells were separated by microbeads, and the CD11b positive cells were incubated with 60 μg of SLAMF5 blocking antibody or with IgG control antibody for 48 h. The cells were stained and analyzed by FACS. Dead cells were excluded from analysis by Zombie Live/Dead staining. Bar graphs show: (D) The expression of MHCII in % in primary CD45+ CD11b+ myeloid cells. (E) The expression of CD80 in % in primary CD45+ CD11b+ myeloid cells. (F) The expression of CD52 in % in primary CD45+ CD11b+ myeloid cells (SLAMF5-blocking $n = 10$ IgG $n = 10$). Graphs show three independent repeats. Two-tailed unpaired Student $t$ test with 95% confidence levels. **(G–L)** The peripheral blood mononuclear cells of MS patients and healthy controls were separated from the serum and red blood cells by Ficoll. The cells were incubated for 48 h with 60 μg of the SLAMF5 blocking antibody or with the IgG control antibody. The cells were stained and analyzed by FACS. Dead cells were excluded from analysis by Zombie Live/Dead staining. Representative histograms and bar graphs show: (G, H) The expression of HLA-DR in CD3− CD19− CD16+ CD14+ (waterfall gate) monocytes. (I, J) The expression of CD80 in CD3− CD19− CD16+ CD14+ monocytes (Healthy $n = 6$; MS $n = 10$). Graphs show six independent repeats. Representative histogram and bar graph show: (K, L) The expression of CD52 represented as x-fold of treatment compared to IgG-control in CD3− CD19− CD16+ CD14+ monocyte cells (Healthy $n = 3$; MS $n = 4$). Graphs show three independent repeats. Two-tailed unpaired Student $t$ test with 95% confidence levels. **(M–T)** Human iPSC-derived microglia cells were seeded and activated with LPS for 48 h. After LPS induction, the cells were incubated for 48 h with 60 μg of SLAMF5 antibody or the IgG control antibodies. The cells were stained and analyzed by FACS. Dead cells were excluded from analysis by Zombie Live/Dead staining. Representative histograms and bar graphs show: (M, N) the expression of SLAMF5 in CD11b+ cells. (O, P) The expression of HLA-DR in CD11b+ cells. (Q, R) The expression of CD80 in CD11b+ cells. (S, T) The expression of CD52 in CD11b+ cells (SLAMF5-blocking n = 9; IgG $n = 8$). Graphs show three independent repeats. Ordinary one-way ANOVA with Dunnett multiple comparison tests (*$P < 0.05$, **$P < 0.01$, ***$P < 0.001$, ****$P < 0.0001$). Data are shown as mean ± SD. The data underlying this figure can be found in S1 Data.

incubated with either SLAMF5-blocking or isotype control antibodies. Baseline analysis showed that HLA-DR expression was elevated in monocytes from MS patients compared to controls, while CD80 levels did not differ significantly between the two groups (Fig 7G–7J). Notably, SLAMF5 blockade reduced the expression of HLA-DR (Fig 7G and 7H) and CD80 (Fig 7I and 7J) in both MS and control-derived monocytes. Interestingly, CD52 expression was selectively reduced in monocytes from MS patients treated with the SLAMF5-blocking antibody (Fig 7K and 7L), suggesting regulatory effects on these cells.

To further explore SLAMF5's role in human CNS myeloid cells, we utilized human iPSC-derived microglia [39,40]. These cells were treated with LPS to simulate a pro-inflammatory environment, followed by incubation with SLAMF5-blocking or isotype control antibodies. SLAMF5 expression was downregulated in antibody-treated cells (Fig 7M and 7N), and the expression of HLA-DR (Fig 7O and 7P), CD80 (Fig 7Q and 7R), and CD52 (Fig 7S and 7T) was also significantly reduced.

Together, these findings demonstrate that SLAMF5 regulates the activation of both peripheral and CNS myeloid cells in mice and humans. The consistent downregulation of activation markers following SLAMF5 blockade, particularly in MS patient-derived cells, supports the potential of SLAMF5 as a therapeutic target for modulating pathogenic myeloid cell responses in neuroinflammatory diseases such as MS.

## Discussion

Myeloid cells—including monocytes, macrophages, and microglia—play a central role in the immune response to CNS damage in MS. These cells are recruited to sites of injury, where they release pro-inflammatory cytokines, chemokines, and reactive oxygen species that exacerbate neuroinflammation and tissue damage. In addition to their pro-inflammatory role, myeloid cells contribute to the clearance of cellular debris and support tissue repair. However, in MS, their function can become dysregulated, promoting chronic inflammation and neurodegeneration [9,10]. As such, myeloid cells are implicated in both protective and pathogenic processes in MS.

SLAMF5 is a cell surface receptor expressed on immune cells and involved in regulating immune responses. Our previous work demonstrated that SLAMF5 deficiency or antibody-mediated blockade delays disease onset and improves clinical outcomes in EAE, an animal model of MS, by modulating regulatory B cell function via the transcription factor c-MAF [36]. In the present study, we explored the role of SLAMF5 in myeloid cells within both the CNS and peripheral compartments. Our findings show that disruption of SLAMF5 homophilic interactions—either by genetic deficiency or antibody blockade—reduces myeloid cell activation in a CD52-dependent manner.

CD52 is a GPI-anchored glycoprotein expressed on various immune cell types, including B cells, T cells, and monocytes. Although CD52 has been implicated in MS pathogenesis [37], its role in myeloid cells remains poorly defined [63]. The therapeutic anti-CD52 antibody alemtuzumab has demonstrated clinical efficacy in MS by depleting CD52-expressing lymphocytes and dampening immune responses [59]. Our data suggest that SLAMF5 regulates CD52 expression in myeloid cells during inflammation. In EAE-induced mice, CD52—along with MHCII and CD80—is upregulated in activated myeloid cells, and this expression is significantly reduced in the absence of SLAMF5 signaling. Unlike broad immunosuppressive strategies such as alemtuzumab, which deplete both T and B cells and carry risks of infection and secondary autoimmunity SLAMF5 blockade selectively modulates CNS myeloid cell activity and may preserve peripheral immune function, offering a more targeted therapeutic approach.

Mechanistically, we show that SLAMF5 regulates the expression of the transcription factor BHLHE40, which plays a pivotal role in neuroinflammation and EAE progression by controlling cytokine production, including GM-CSF and IL-10 in T cells [61]. In CNS myeloid cells, SLAMF5 promotes BHLHE40 expression and its binding to the CD52 promoter, resulting in elevated CD52 expression. SLAMF5 blockade inhibits this interaction, reducing CD52 levels and leading to decreased myeloid cell activation (S12 Fig).

Our data further suggest that SLAMF5 influences the ability of CNS myeloid cells to support T cell activation. In both full and conditional SLAMF5 knockout models, we observed a significant reduction in the accumulation of CNS lymphocytes and decreased IFN-γ production by CD4$^+$T cells, indicating that SLAMF5-dependent myeloid activation enhances T cell responses during neuroinflammation.

While the full SLAMF5 knockout model provides valuable insights into the receptor's role in immune regulation, certain limitations must be acknowledged. Global deletion of SLAMF5 throughout development and across all immune cell types may induce compensatory mechanisms or systemic alterations beyond the myeloid compartment, potentially confounding the interpretation of cell type–specific phenotypes. To address this limitation, we repeated key experiments using a cKO model targeting CX3CR1$^+$cells. This myeloid-specific deletion yielded phenotypic and mechanistic outcomes consistent with those observed in the full knockout model, reinforcing the conclusion that SLAMF5 plays a critical and intrinsic role within the myeloid lineage.

Importantly, we extended our findings to human systems. SLAMF5 blockade reduced the expression of HLA-DR, CD80, and CD52 in monocytes derived from the peripheral blood of MS patients, as well as in LPS-activated human iPSC-derived microglia. These results confirm a conserved role for SLAMF5 in regulating human myeloid cell activation and support its potential as a therapeutic target in MS.

In conclusion, our study identifies SLAMF5 as a central regulator of myeloid cell activation via a BHLHE40–CD52 signaling axis. Targeting SLAMF5 offers a novel and selective approach to modulating neuroinflammation in MS, potentially limiting CNS damage while minimizing systemic immunosuppression.

## Supporting information

**S1 Fig. Mice lacking SLAMF5 exhibited delayed disease onset and milder motor dysfunction.** EAE (MOG35-55) was induced in WT and in SLAMF5-deficient mice. **(A)** Daily Mean clinical scoring of the disease. **(B)** Bar graph shows the Area Under the Curve of the clinical score. Mann-Whitney test (EAE induced WT group, $n = 52$; EAE-induced SLAMF5−/− group, $n = 38$). (*$P < 0.05$, **$P < 0.01$, ***$P < 0.001$, ****$P < 0.0001$). Data are shown as mean ± SD. The data underlying this figure can be found in S1 Data.
(TIFF)

**S2 Fig. SLAMF5 expression specifically in myeloid cells is correlated to the disease severity in the EAE model.** EAE (MOG35-55) was induced in WT and in SLAMF5-deficient mice. At day 15, the mice were sacrificed and their brains were excised. Their brains were processed, and immune cells were isolated, stained and analyzed by FACS. Dead cells were excluded from analysis by Zombie Live/Dead staining. Bar graphs showing **(A)** Correlation between SLAMF5 expression in CD45**+** CD11b+ myeloid cells in the brain and disease severity($n = 18$). **(B)** Percent expression of SLAMF5 in the CD45**+** CD11b− non-myeloid cells in the brain (Healthy WT $n = 8$; EAE WT $n = 10$). **(C)** Correlation between the SLAMF5 expression in CD45**+** CD11b− non-myeloid cells in the brain and disease severity ($n = 18$). Graphs show three independent determinations. Two-tailed unpaired Student $t$ test with 95% confidence levels. (*$P < 0.05$, **$P < 0.01$, ***$P < 0.001$, ****$P < 0.0001$). Data are shown as mean ± SD. The data underlying this figure can be found in S1 Data.
(TIFF)

**S3 Fig. MHCII and CD80 expression of myeloid cells correlate with disease severity in the EAE model.** EAE (MOG35-55) was induced in WT and in SLAMF5-deficient mice. On day 15, the mice were sacrificed and their brains were excised. Their brains were processed and immune cells were isolated, stained and analyzed by FACS. Dead cells were excluded from analysis by Zombie Live/Dead staining. Graphs show: **(A)** Correlation between MHCII expression in CD45**+** CD11b+ myeloid cells in the brain and disease severity ($n = 12$). **(B)** Correlation between the CD80 expression in CD45**+** CD11b+ myeloid cells in the brain and disease severity ($n = 12$). Graphs show three independent determinations. The data underlying this figure can be found in S1 Data.
(TIFF)

**S4 Fig. Gating of the total immune population and MHCII and CD80 expression in TMEM119+ microglial cells.** EAE (MOG35-55) was induced in mice. At day 15, the mice were sacrificed, and their brains were excised. The brains were processed and immune cells were isolated, stained and analysed by FACS. Dead cells were excluded from analysis by Zombie Live/Dead staining. **(A)** Representative plot showing gating strategy for brain suspension cells: CD45 was used to identify immune cells, CD11b for myeloid cells, and TMEM119+ for a specific population of microglial cells. Bar graphs show **(B)** the expression of MHCII in CD45**+** CD11b+ TMEM119**+** microglial cells (WT $n = 20$; SLAMF5−/− $n = 25$). **(C)** the expression of CD80 in CD45**+** CD11b+ TMEM119**+** microglial cells (WT $n = 22$; SLAMF5−/− $n = 27$). Two-tailed unpaired Student $t$ test with 95% confidence levels. (*$P < 0.05$, **$P < 0.01$, ***$P < 0.001$, ****$P < 0.0001$). Data are shown as mean ± SD. The data underlying this figure can be found in S1 Data.
(TIFF)

**S5 Fig. The conditional knock out of SLAMF5 in brain myeloid cells do not affect the activation of myeloid cells in the spleen.** EAE (MOG35-55) was induced in control and CX3CR1 cre SLAMF5 flox mice. At day 15, the mice were sacrificed, and their spleens were excised. The spleens were processed, and immune cells were isolated, stained and analyzed by FACS. Dead cells were excluded from analysis by Zombie Live/Dead staining. Bar graphs show **(A)** the expression of MHCII in F4/80**+** CD11b+ myeloid cells (control EAE $n = 13$; cKO SLAMF5 EAE −/− $n = 10$). **(B)** the expression of CD80 in F4/80**+** **C**D11b+ myeloid cells (control EAE $n = 13$; cKO SLAMF5 EAE −/− $n = 10$). Two-tailed unpaired Student $t$ test with 95% confidence levels. (*$P < 0.05$, **$P < 0.01$, ***$P < 0.001$, ****$P < 0.0001$). Data are shown as mean ± SD. The data underlying this figure can be found in S1 Data.
(TIFF)

**S6 Fig. Blocking SLAMF5 significantly delayed the onset of the disease and reduced its severity.** EAE (MOG35-55) was induced in WT mice and mice were treated with 3 injections, I.V, at days 7,9, and 12 postinduction with 150 µg of the SLAMF5 blocking antibody or the IgG control. Mice were monitored daily for 15 days. **(A)** Graph shows the Daily Mean clinical scoring of the disease. **(B)** Bar graph shows the Area Under the Curve of the clinical score. (SLAMF5-blocking $n = 22$; IgG $n = 26$). Mann–Whitney test. (*$P < 0.05$, **$P < 0.01$, ***$P < 0.001$, ****$P < 0.0001$). Data are shown as mean ± SD. The data underlying this figure can be found in S1 Data.
(TIFF)

**S7 Fig. mRNA level of CD52, IRF7, DDX58, ITGAX, VCAM1, IRF1, STAT1, BHLEH40, APOE, and HSPA1A in brain myeloid cells of WT Healthy, WT EAE, and SLAMF5−/− EAE mice.** EAE (MOG35-55) was induced in WT and SLAMF5−/− mice. At the peak of the disease, the mice were sacrificed, and their brains were excised. The brains were processed, and immune cells were isolated, stained for dead cells, and labeled with anti-CD45 and anti-CD11b antibodies. The double positive population was sorted. RNA was isolated and cDNA was synthesized. The expression levels of genes were determined by qRT-PCR. Data are presented as fold change relative to the WT EAE group. Graphs show two independent determinations. Ordinary one-way ANOVA with Dunnett multiple comparison tests. (*$P < 0.05$, **$P < 0.01$, ***$P < 0.001$, ****$P < 0.0001$). Data are shown as mean ± SD. The data underlying this figure can be found in S1 Data.
(TIFF)

**S8 Fig. mRNA level of CD52 and BHLHE40 in N9 microglial cells after siRNA electroporation. (A)** N9, murine microglia cell line cells, were electroporated with CD52 siRNA or Control siRNA. RNA was isolated and cDNA was synthesized. The expression levels of CD52 were determined by qRT-PCR. Bar graph showing mRNA level of CD52 in control siRNA ($n = 5$), CD52 siRNA ($n = 5$). Graphs show two independent determinations. **(B)** N9, murine microglia cell line cells, were electroporated with BHLHE40 siRNA or Control siRNA. RNA was isolated and cDNA was synthesized. The expression levels of BHLHE40 were determined by qRTPCR. Bar graph showing mRNA level of BHLHE40 in control siRNA ($n = 4$),

BHLHE40 siRNA ($n=4$). Graphs show two independent determinations. Data are presented as fold change relative to the siRNA control group, which was normalized to a value of 1. Two-tailed unpaired Student $t$ test with 95% confidence levels. (*$P<0.05$, **$P<0.01$, ***$P<0.001$, ****$P<0.0001$). Data are shown as mean±SD. The data underlying this figure can be found in S1 Data.
(TIFF)

**S9 Fig. Binding of BHLHE40 in the promotor area of TMEM119 and IL10RA in brain myeloid cells treated with SLAMF5 blocking or IgG control.** EAE (MOG35-55) was induced in WT mice. At the peak of the disease, the mice were sacrificed, and their brains were excised. The brains were processed, and immune cells were isolated, stained for dead cells, and labeled with anti-CD45 and anti-CD11b antibodies. The double positive population was sorted. The sorted cells were incubated with SLAMF5 blocking antibody or with the isotype control for 1 h. Then, cells were harvested, and ChIP was performed. Binding of BHLHE40 to the promoter area of the TMEM119 and IL10RA genes were determined by RT-qPCR. Graph presents the percent enrichment of the input. Two-tailed unpaired Student $t$ test with 95% confidence levels. (*$P<0.05$, **$P<0.01$, ***$P<0.001$, ****$P<0.0001$). Data are shown as mean±SD. The data underlying this figure can be found in S1 Data.
(TIFF)

**S10 Fig. The conditional knock out of SLAMF5 in brain myeloid cells do not affect the activation of CD52 in myeloid cells in the spleen.** EAE (MOG35-55) was induced in control and CX3CR1 cre SLAMF5 flox mice. At day 15, the mice were sacrificed, and their spleens were excised. The spleens were processed, and immune cells were isolated, stained and analyzed by FACS. Dead cells were excluded from analysis by Zombie Live/Dead staining. Bar graph shows the expression of CD52 in F4/80+ CD11b+ myeloid cells (control EAE $n=13$; cKO SLAMF5 EAE −/− $n=10$). Two-tailed unpaired Student $t$ test with 95% confidence levels. (*$P<0.05$, **$P<0.01$, ***$P<0.001$, ****$P<0.0001$). Data are shown as mean±SD. The data underlying this figure can be found in S1 Data.
(TIFF)

**S11 Fig. CD80 expression on spleen myeloid cells is correlated to disease severity in an EAE model.** EAE (MOG35-55) was induced in WT and in SLAMF5-deficient mice. On day 15, the mice were sacrificed and their spleens were excised. Immune cells were isolated, stained and analyzed by FACS. Dead cells were excluded from analysis by Zombie Live/Dead staining. Representative bar graph shows the correlation between the CD80 expression on CD11b+ F4/80 myeloid cells in the spleen and the disease severity ($n=14$). Graphs show four independent determinations. The data underlying this figure can be found in S1 Data.
(TIFF)

**S12 Fig. Suggested model.** *Using Biorender.com.*
(TIFF)

**S1 Table. Mouse antibodies.**
(TIFF)

**S2 Table. Human antibodies.**
(TIFF)

**S3 Table. PCR Primers.**
(TIFF)

**S1 Data. Raw experimental measurements for the generation of Figs 1–7 and Supplementary Figs S1 –11.**
(XLSX)

## Acknowledgments

The authors wish to thank members of the Shachar lab for fruitful discussion and support. I.S. is the incumbent of the Dr. Morton and Ann Kleiman Professorial Chair. M.T. is the incumbent of the Carolito Stiftung Research Fellow Chair in Neurodegenerative Diseases. Sapir Mark is acknowledged for valuable assistance in preparing the histological samples.

Sincere gratitude is extended to Dr. Sacha Lebon for his support and precious help throughout this work.

## Author contributions

**Conceptualization:** Laura Bellassen, Idit Shachar.

**Data curation:** Laura Bellassen, Keren David, Bar Lampert, Avital Sarusi-Portuguez, Jazz Lubliner, Shirly Becker-Herman.

**Formal analysis:** Laura Bellassen, Keren David, Bar Lampert, Avital Sarusi-Portuguez, Jazz Lubliner, Ori Brenner.

**Funding acquisition:** Idit Shachar.

**Investigation:** Laura Bellassen, Idit Shachar.

**Methodology:** Laura Bellassen, Keren David, Bar Lampert, Michael Tsoory, Jazz Lubliner, Eran Hornstein.

**Project administration:** Shirly Becker-Herman, Idit Shachar.

**Resources:** Michael Osherov, Ron Milo.

**Writing – original draft:** Laura Bellassen, Idit Shachar.

**Writing – review & editing:** Eran Hornstein, Ron Milo.

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
