## [Editor Report · Decision Letter 0]

6 Jun 2025

Dear Idit, 

Thank you for submitting your manuscript entitled "SLAMF5 Regulates Myeloid-Cell Mediated Neuroinflammation in Multiple Sclerosis" for consideration as a Research Article by PLOS Biology.

Your manuscript has now been evaluated by the PLOS Biology editorial staff, as well as by the previous academic editor, and I am writing to let you know that we would like to send your submission back to the previous reviewers.

Once your full submission is complete, your paper will undergo a series of checks in preparation for peer review. After your manuscript has passed the checks it will be sent out for review. To provide the metadata for your submission, please Login to Editorial Manager (https://www.editorialmanager.com/pbiology) within two working days, i.e. by Jun 08 2025 11:59PM.

Kind regards,

Melissa

Melissa Vazquez Hernandez, Ph.D.

Associate Editor

PLOS Biology

---

## [Decision Letter · Decision Letter 1]

13 Jul 2025

Dear Idit,

Thank you for your patience while we considered your revised manuscript "SLAMF5 Regulates Myeloid-Cell Mediated Neuroinflammation in Multiple Sclerosis" for consideration as a Research Article at PLOS Biology. Your revised study has now been evaluated by the PLOS Biology editors, the Academic Editor and two of the original reviewers.

As you will see in the reports, the reviewers recognize the good job done addressing their previous concerns, but Reviewers #1 still would like some aspects to be addressed. Specifically, Reviewer #1 is concerned that key mechanistic differences between the global KO and conditional KO models were not addressed, and that claims about SLAMF5’s role in myeloid cell–mediated neuroinflammation remain descriptive without functional validation, and thinks cell-type specific analyses are required to support the conclusions. The rationale for using global KO instead of cKO for transcriptomics should be clearly justified and discussed as a limitation. The reviewer also requests clarification on whether histopathology was quantified and that statistical comparisons between naïve and EAE controls be included.

In light of the reviews, which you will find at the end of this email, we are pleased to offer you the opportunity to address the remaining points from the reviewers in a revision that we anticipate should not take you very long. We will then assess your revised manuscript and your response to the reviewers' comments with our Academic Editor aiming to avoid further rounds of peer-review, although we might need to consult with the reviewers, depending on the nature of the revisions.

**IMPORTANT - SUBMITTING YOUR REVISION**

*Resubmission Checklist*

*Published Peer Review*

*PLOS Data Policy*

*Blot and Gel Data Policy*

Sincerely,

Melissa

Melissa Vazquez Hernandez, Ph.D.

Associate Editor

PLOS Biology

REVIEWERS' COMMENTS.

Reviewer #1: 

Reviewer 1 comments from point-by-point response

Reviewer comment 1: 

While the authors may consider there is "no overlap between these two studies"; two reviewers, myself and Reviewer 3, independently noted the data similarity between the authors prior study in Nat Commun.

"we are unclear as to why the reviewer perceives a mechanistic difference between the full knockout (KO) and the conditional knockout (cKO) models. In both cases, SLAMF5 expression is effectively downregulated in sorted microglial cells, and the comparisons are made between cells that express or do not express SLAMF5."

The authors title and text claim a specific role for SLAMF5 in regulating myeloid-cell mediated neuroinflammation. To support this conclusion, assays using cell-type specific ablation are critical to inform on cell autonomous functions. The authors data in the global SLAMF5 KO (original Fig 1A-B) showed that animals begin to show EAE clinical symptoms by day 14. The cKO mice are largely (80% of the mice in Fig. 2H,I) protected from EAE disease based on the curve and AUC of zero. Based on the scores, the data suggest that myeloid-specific ablation has a unique difference than global KOs in EAE pathogenesis. In KO mice, given the authors performed flow and RNA-seq experiments on all CD45+CD11b+ cells, which in EAE includes many cell-types and activation states of each of them, cell-specific analysis is required to support such conclusions. How Slamf5 regulates myeloid cell mediated neuroinflammation in EAE remains unclear. In microglia or macrophages, is there any signaling pathways deregulated in Slamf5 dependent manner? The RNA-seq data suggest some, but functional validation studies are not explored, and thus the study remains descriptive as presented. 

The Cx3cr1-CreERT2 strain is widely used for multi-omic studies including the EAE model, one example here (Ennerfelt et al Cell 2022). The rationale to select global knockout strain over cKO approach "to ensure maximal and uniform deletion for transcriptomic analysis" should be addressed as a caveat in discussion.

Histopathology: I commend the authors for their new experiment and inclusion of histological data from the control and cKO groups. The authors state "We performed spinal cord histology on EAE-induced cKO and control mice, which revealed significantly reduced lymphocyte infiltration in the cKO animals". Did the authors perform image quantification?

Naive State Comparison: Statistics between naïve-control and Contorl EAE are not included in Fig 2K, M.

Reviewer comment 2:

What role do the authors suggest microglial MHC-II plays in relation to SLAMF5 in EAE? It has been reported microglial MHCII is dispensable during EAE (Wolf et al Eur J Immunol 2018). Since the authors do not distinguish between microglia vs infiltrating myeloid cells in their analyses, how do the authors interpret the contribution of SLAMF5 in microglia vs peripheral myeloid cells?

Reviewer comment 3: RNA-seq data

I still find the RNA-seq dataset difficult to interpret. There is 76% gene variance between 1 WT and the KOs, but it is unclear if this is because the authors sorted all CD45+CD11b+ cells from the inflamed CNS of EAE mice. The myeloid composition, and their activation states (microglial states, macrophages states, neutrophil states etc,) change dramatically from onset to peak disease (examples: Jordao et al Science 2019; Mendiola et al Nat Immunol 2020; Peruzzotti-Jametti et al Nature 2024). The selected genes in Fig. 5A support the reduced gene activation in CD45+CD11b+ cells of cKO brains in EAE.

Additional comments of revised manuscript

1. The authors tend to present flow cytometric histograms of markers but then the quantification is a cell percentage. Is there a reason why the maker MFI is not presented? For example, but throughout manuscript, Fig 1E, F. 

2. The title suggests a causal role of SLAMF5 in neuroinflammation in MS, which is too strong based on the presented data. A title reflecting MS models or CNS inflammation would be more appropriate.

Reviewer #2: 

Multiple Sclerosis (MS) is an inflammatory, demyelinating, and neurodegenerative disease of the central nervous system that affects over 2 million people worldwide and typically begins in young adulthood. Currently, no treatment can halt disease progression, highlighting the urgent need to better understand its underlying mechanisms in order to develop effective preventive strategies.

In this study, the authors investigated the expression and function of Factor SLAMF5, a regulator of immune cell function, using an inflammatory mouse model of MS (EAE). They show that totally or partially blocking SLAMF5 function delayed disease onset and reduced progression in this well-established MS model, suggesting SLAMF5 may serve as a therapeutical target in MS. While this research builds on the authors' previous works on the role of SLAMF5 in immune cells, the new findings presented here are of interest.

In this revised version, the authors have made substantial changes in the manuscript and included newly generated data, that enhance the quality of the work.

Most of my initial questions and concerns regarding this study have been satisfactorily addressed (except maybe the point "p13: how can the ICV with SLAMF5 blocking antibody and the cKO of SLAMF5 have so drastic effects compared to the SLAMF5 mutant regarding EAE scores? This is a major concern regarding the models used in this study.").

Consequently, I do not have any major objections to its publication.

---

## [Editor Report · Decision Letter 2]

10 Aug 2025

Dear Idit,

Thank you for your patience while we considered your revised manuscript "SLAMF5 Regulates Myeloid-Cell Mediated Neuroinflammation in Multiple Sclerosis" for publication as a Research Article at PLOS Biology. This revised version of your manuscript has been evaluated by the PLOS Biology editors, and the Academic Editor.

Based on our Academic Editor's assessment of your revision, we are likely to accept this manuscript for publication. However, the Academic Editor requested that you please discuss the caveats of the full KO in the discussion. Please also make sure to address the following data and other policy-related requests.

a) We routinely suggest changes to titles to ensure maximum accessibility for a broad, non-specialist readership, and to ensure they reflect the contents of the paper. In this case, we would suggest a minor edit to the title, as follows. Please ensure you change both the manuscript file and the online submission system, as they need to match for final acceptance:

"The immune receptor SLAMF5 regulates myeloid-cell mediated neuroinflammation in multiple sclerosis"

b) Please add the grant number in the Financial Disclosure statement in the manuscript details.

c) You state that your study does not require an ethics statement. This is incorrect as you perform animal experiments. I also note that in your subsecion "Mice" in the Materials and Methods secsion you say "All procedures involving animals were approved by the Animal Research Committee of the Weizmann Institute of Science." The Ethics statement needs to be a separate, independent (and the first) subheading in the Material & Methods section. It must include the full name of the IACUC/ethics committee that reviewed and approved the animal care and use, as well as the protocol/permit/project license number. https://journals.plos.org/plosbiology/s/ethical-publishing-practice

Please supply the numerical values either in the a supplementary file or as a permanent DOI’d deposition for the following figures:

Figure 1BDF-J, 2CEGHIKM-Q, 3ABDFGHJ-NPR, 4A-E, 5ABCEGIK, 6BDFHJK, 7GIKMOQS, S1AB, S2ABC, S3AB, S4BC, S5AB, S6AB, S7, S8AB, S9, S10, S11

e) Please cite the location of the data clearly in all relevant main and supplementary Figure legends, e.g. “The data underlying this Figure can be found in S1 Data” or “The data underlying this Figure can be found in https://doi.org/10.5281/zenodo.XXXXX”

f) For figures containing FACS data (Figures 1ACE, 2BDFJL, 3CEGIOQ, 5DFHJL, 6ACEGI, 7A-FHJLNPRT, S4A), please provide the FCS files and a picture showing the successive plots and gates that were applied to the FCS files to generate the figure. We ask that you please deposit this data in the FlowRepository (https://flowrepository.org/) and provide the accession number/URL of the deposition in the Data Availability Statement in the online submission form.

g) Supplementary files (e.g., excel). Please ensure that all data files are uploaded as 'Supporting Information' and are invariably referred to (in the manuscript, figure legends, and the Description field when uploading your files) using the following format verbatim: S1 Data, S2 Data, etc. Multiple panels of a single or even several figures can be included as multiple sheets in one excel file that is saved using exactly the following convention: S1_Data.xlsx (using an underscore).

h) Please add a scale bar in the following microscopy pictures in Figure 2N

i) You mentioned the Data was submitted to GEO. However, it is not clear which data was submitted there. Please also provide the accession code or a reviewer link so that we may view your data before publication. Please ensure that your Data Statement in the submission system accurately describes where your data can be found and is in final format, as it will be published as written there. I would also like to point out that in your text, there is no statement of Data Availability, please provide it. 

j) Per journal policy, if you have generated any custom code during the course of this investigation, please make it available without restrictions. Please ensure that the code is sufficiently well documented and reusable, and that your Data Statement in the Editorial Manager submission system accurately describes where your code can be found. 

We expect to receive your revised manuscript within two weeks. 

*Published Peer Review History*

*Press*

Sincerely,

Melissa

Melissa Vazquez Hernandez, Ph.D.

Associate Editor

PLOS Biology

---

## [Editor Report · Decision Letter 3]

19 Aug 2025

Dear Idit,

Thank you for the submission of your revised Research Article "The immune receptor SLAMF5 regulates myeloid-cell mediated neuroinflammation in multiple sclerosis" for publication in PLOS Biology. On behalf of my colleagues and the Academic Editor, Dr. Richard Daneman, I am pleased to say that we can in principle accept your manuscript for publication, provided you address any remaining formatting and reporting issues. These will be detailed in an email you should receive within 2-3 business days from our colleagues in the journal operations team; no action is required from you until then. Please note that we will not be able to formally accept your manuscript and schedule it for publication until you have completed any requested changes.

IMPORTANT: Thank you for following all our previous requests. However, we do require that you provide all fcs files for the FACS data present in Figures 1ACE, 2BDFJL, 3CEGIOQ, 5DFHJL, 6ACEGI, 7A-FHJLNPRT, S4A. Since FlowRepository is not available, please do so by uploading the files to Zenodo or by uploading them to our system, and provide this URL in the manuscript and Data Availability Statement. I have asked my colleagues to include this request alongside their own. Thank you!

PRESS

Sincerely, 

Melissa

Melissa Vazquez Hernandez, Ph.D., Ph.D.

Associate Editor

PLOS Biology
